# Unscheduled DNA replication in G1 causes genome instability and damage signatures indicative of replication collisions

Karl-Uwe Reusswig [1,4,5], Julia Bittmann[1], Martina Peritore[1,6], Mathilde Courtes [2], Benjamin Pardo [2], Michael Wierer [3,7], Matthias Mann [3] & Boris Pfander [1,6,8] ✉

DNA replicates once per cell cycle. Interfering with the regulation of DNA replication initiation generates genome instability through over-replication and has been linked to early stages of cancer development. Here, we engineer genetic systems in budding yeast to induce unscheduled replication in a G1-like cell cycle state. Unscheduled G1 replication initiates at canonical S-phase origins. We quantify the composition of replisomes in G1- and S-phase and identified firing factors, polymerase α, and histone supply as factors that limit replication outside S-phase. G1 replication per se does not trigger cellular checkpoints. Subsequent replication during S-phase, however, results in over-replication and leads to chromosome breaks and chromosome-wide, strand-biased occurrence of RPA-bound single-stranded DNA, indicating head-to-tail replication collisions as a key mechanism generating genome instability upon G1 replication. Low-level, sporadic induction of G1 replication induces an identical response, indicating findings from synthetic systems are applicable to naturally occurring scenarios of unscheduled replication initiation.

To ensure that DNA is replicated precisely once per cell cycle, eukaryotic DNA replication initiation involves two steps, with each being restricted to different cell cycle phases[1]. In the first step (origin licensing), replicative helicase precursors are loaded at origins of replication[2]; in the second step (origin firing), cyclin-dependent kinase (CDK) and Dbf4-dependent kinase (DDK) promote helicase activation by facilitating the association of helicase accessory factors[3–5]. Multiple regulatory mechanisms ensure temporal separation of the two steps. Experimentally inducing unscheduled helicase loading or activation

requires engineering synthetic bypasses to these endogenous controls and results in over-replication[6–10].

Studies in budding yeast achieved over-replication via unscheduled helicase loading in the S/M-phase of the cell cycle (so-called re-replication)[11]. Specifically, these experimental systems override CDK controls to helicase loading to promote helicase re-loading at replication origins that were already activated[6,7,12]. Replication collisions and double-strand breaks (DSBs) occur as a consequence of this over-replication[13,14] and are accompanied by hallmarks of genome instability

[1]DNA Replication and Genome Integrity, Max Planck Institute of Biochemistry, 82152 Martinsried, Germany. [2]Institut de Génétique Humaine (IGH), Université de Montpellier – Centre National de la Recherche Scientifique, 34396 Montpellier, France. [3]Proteomics and Signal Transduction, Max Planck Institute of Biochemistry, 82152 Martinsried, Germany. [4]Present address: Department of Cell Biology, Blavatnik Institute, Harvard Medical School, Boston, MA 02115, USA. [5]Present address: Department of Pediatric Oncology, Dana-Farber Cancer Institute, Boston, MA 02215, USA. [6]Present address: Genome Maintenance Mechanisms in Health and Disease, Institute of Aerospace Medicine, German Aerospace Center (DLR), 51147 Cologne, Germany. [7]Present address: Proteomics Research Infrastructure, University of Copenhagen, 2200 Copenhagen, Denmark. [8]Present address: Genome Maintenance Mechanisms in Health and Disease, Institute of Genome Stability in Ageing and Disease, CECAD Research Center, University of Cologne, 50931 Cologne, Germany. ✉e-mail: bpfander@biochem.mpg.de

including gene amplifications, gross chromosomal rearrangements (GCRs), and aneuploidy[6,7,11,13–18].

In metazoans, additional CDK-independent mechanisms control helicase loading and interfering with these mechanisms also causes over-replication[19–22]. For example, unscheduled helicase loading results in extensive DNA damage and loss of cellular viability in cultured human cells and *Xenopus laevis* egg extracts[23–31]. In *Drosophila melanogaster*, follicle cells undergo developmentally programmed over-replication at specific genomic loci and DSBs occur at sites of potential head-to-tail replication collisions[32–35]. Thus, both natural and synthetic over-replication systems appear to generate DSBs and one can speculate that some form of fork stalling or collision is involved in generating this damage.

Alternatively, over-replication can also be caused by unscheduled helicase activation in G1 followed by replication in S-phase. To trigger unscheduled G1 replication, a system needs to override the cell cycle control of helicase activation. This is currently only possible in budding yeast, where the minimal set of CDK and DDK targets for helicase activation has been identified[36–39]. Previous work indicated that unscheduled helicase activation results in genome instability, aneuploidy, and cell death[38], but little is known about how cells respond to re-replicated DNA in G1, what consequences this has for the following S-phase, and what kind of DNA structures are generated. Furthermore, it is unknown if there are other mechanisms constraining unscheduled replication in G1 besides the requirement for the activity of CDK and DDK.

Genome instability caused by over-replication has been linked to early stages of cancer development[40–43]. Common oncogenic drivers such as overexpression of cyclin E or MYC deregulate the G1/S transition and cause replication stress[44–46]. Unscheduled helicase activation in G1 or at the G1/S transition could contribute significantly to the replication stress, but we currently lack sensitive methodology to detect it. It is therefore unknown whether unscheduled replication occurs upon de-regulation of the G1/S transition by oncogenes. A marker for unscheduled replication would thus facilitate the detection and investigation of early stages of cancer development.

Here, we engineered different genetic systems in budding yeast to induce unscheduled helicase activation and thereby replication in a G1-like state and investigated its characteristics, constraints, and consequences. We found that unscheduled G1 replication initiates at canonical origins on all chromosomes but progressed slower than canonical S-phase replication. Quantitative proteomics revealed a reduced number of replisomes but also differences in replisome composition compared to S-phase replication. Testing for factors that constrain G1 replication, we found histone availability to be limiting, suggesting that histone supply is a crucial bottle-neck for replisome progression. Importantly, when we investigated the consequences of unscheduled G1 replication, we found that subsequent S-phase replication strongly aggravated genome instability. Specifically, we observed chromosome breaks and DNA damage checkpoint activation after release into S-phase. These phenotypes were completely suppressed when further replication initiation in S-phase was blocked, indicating that successive rounds of replication caused the observed DNA damage. Data from strand-specific ChIP-sequencing of RPA-bound single-stranded DNA revealed a characteristic pattern of strand-biased RPA accumulation along whole chromosomes. The pattern indicates that single-ended DSBs were generated by head-to-tail replication collisions and in turn expose single-stranded DNA. Using a complementary strategy, we induced low levels of sporadic G1 replication and observed a similar cellular response indicating that our engineered systems reveal insights of physiological significance and that asymmetric accumulation of RPA-bound single-stranded DNA is a highly sensitive marker of acute over-replication.

## Results

### Unscheduled G1 replication initiates at canonical replication origins

To engineer a system able to initiate unscheduled replication in a G1-like cell cycle state, we adapted previously published strategies that allow the bypass of CDK and DDK control of replication[36,37,47]. In order to minimally interfere with cellular physiology, we implemented conditional expression of replication initiation proteins from galactose-inducible promoters (Fig. 1a). Expression of high levels of Dpb11 together with a CDK phosphorylation-mimicking allele of *SLD2* (*sld2-T84D*) generates a bypass to CDK-regulation of replication initiation and additional expression of the cell cycle-regulated DDK subunit Dbf4 allowed DDK activation in G1[48–50].

Using these systems, we first arrested cells in G1 using the mating pheromone α-factor and induced DNA replication by bypassing CDK and DDK controls of DNA replication. We followed DNA synthesis by flow cytometry, measuring either incorporation of the nucleoside analog EdU during DNA synthesis or the increase in total cellular DNA (Fig. 1b). We observed a linear increase in DNA content over time, resulting in a 47% increase in average DNA content 5 h after induction with CDK bypass and in a 78% increase with CDK/DDK bypass compared to control cells that showed low levels of DNA synthesis arising from cell cycle-independent mitochondrial DNA replication (Supplementary Fig. 1a). In all conditions, changes in total DNA correlated with changes in EdU-labeled DNA, indicating newly synthesized DNA was quantitatively labeled (Supplementary Fig. 1b). Importantly, cells remained in a G1-like state after bypass of CDK and DDK regulation of replication initiation as determined by principal component analysis of transcriptomes via RNA-seq (Supplementary Fig. 1c–e). These systems therefore allow the synthetic induction of DNA synthesis during G1-arrest and we therefore refer to this DNA replication in a G1-like state as G1 replication hereafter. Thus, both systems can be used to trigger unscheduled replication in G1 and tune the level of G1 replication.

To assess if unscheduled replication in G1 occurs genome-wide, we induced both systems for 3 h in G1-arrested cells and purified EdU-labeled DNA replication products via biotin handles for next-generation sequencing. Replication products were observed from all chromosomes, albeit to different extents (Fig. 1c): CDK bypass enriched for products from chromosomal regions that also replicate early in a regular S-phase (Fig. 1c middle, Supplementary Fig. 1f middle), whereas bypass of both CDK and DDK regulation resulted in a more even coverage of all chromosomes (Fig. 1c bottom, Supplementary Fig. 1f right). Late replicating regions, however, particularly those close to telomeres were still underrepresented in these samples Thus, unscheduled replication in G1 occurs in both systems to different extents on all chromosomes and appears to follow the same relative timing as replication in S-phase.

To determine if unscheduled G1 replication initiates from canonical replication origins, we limited DNA synthesis by addition of 60 mM hydroxyurea (HU) and purified EdU-labeled DNA replication products for next-generation sequencing (Fig. 1d, e). We detected replication initiation at origins which fire early in S-phase[51], as indicated by symmetrical peaks around these loci. Most early origins were used in the CDK/DDK bypass conditions, while DNA replication initiated only from a subset in the CDK bypass conditions (Fig. 1d, e). This finding is consistent with replication occurring more evenly across chromosomes in the CDK/DDK bypass conditions (Fig. 1c). Thus, unscheduled replication in G1 initiates from the same canonical replication origins as in S-phase. Furthermore, the differences between the CDK and CDK/DDK bypass indicate that CDK and DDK activation collectively leads to replication initiation from early-firing origins, while with limited DDK activation only a subset of these origins becomes active.

ᴬ

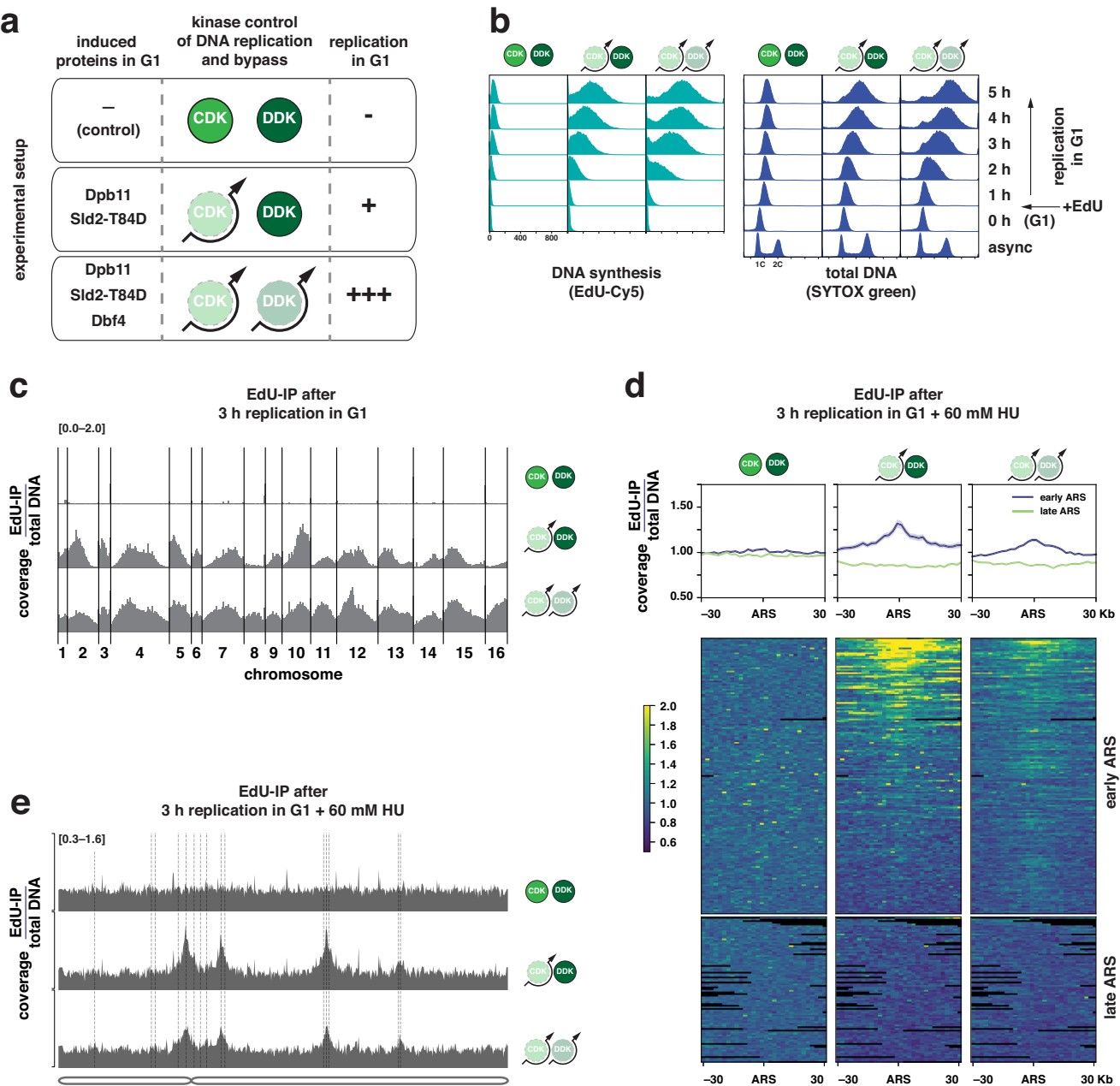

**Fig. 1 | Unscheduled replication in G1 initiates at canonical replication origins genome-wide. a** Summary of engineered genetic changes that allow to bypass cell cycle control of DNA replication initiation and induce unscheduled replication in G1. Indicated proteins and variants are expressed from a galactose-inducible *pGAL1-10* promoter. Experimental setup for G1 replication involves G1 cell cycle arrest using α-factor in raffinose medium, followed by induction of G1 replication by addition of galactose. Expression of Dpb11 and Sld2-T84D allows for bypass of CDK controls, additional expression of Dbf4 allows for bypass of DDK controls. Pictograms indicate genetic bypass in all figures. **b** Bypassing CDK control and CDK/DDK control generates different levels of unscheduled replication in G1. Cells were arrested in G1 and replication was induced by adding galactose. EdU (100 μM) was added to the G1-arrested cells right after induction of replication. Cells were harvested at indicated timepoints after replication induction and EdU-containing DNA was labeled with Cy5. Total DNA content (stained by SYTOX green) and newly synthesized DNA (EdU-Cy5-labeled) were measured by flow cytometry. Data are representative of *n* = 3 biological replicates. **c** Unscheduled replication in G1 after

bypass of CDK or CDK/DDK control occurs genome-wide. Experiment as in (**b**), but EdU-labeled DNA as a proxy for DNA synthesis was isolated after 3 h of G1 replication and sequenced. Sequencing reads were mapped to the *S. cerevisiae* genome and normalized for input total DNA. Data are representative of *n* = 2 biological replicates. **d** and **e** Unscheduled G1 replication initiates at canonical replication origins. Experiment as in (**b**)/(**c**), but 60 mM hydroxyurea (HU) was added to the medium when replication was induced. **d** G1 replication initiates at early-firing replication origins (autonomous replicating sequences (ARS)). Input-normalized coverage of 60 kb windows shows EdU-labeled replication products around ARS after 3 h replication in G1 at ARS firing either early (blue) or late (green) in S-phase. (top) Summarizing profile plots of mean coverage (dark) ± SE (light) at the ARSs ± 30 Kb. (bottom) Heatmaps with 2 Kb bin size, each row corresponds to an individual ARS. Data are representative of *n* = 2 biological replicates. **e** Representative example of input-normalized traces for EdU-containing DNA spanning the entire chromosome 4. The scale of the *y*-axis (log) is given in the top-left corner. Dotted lines indicate early-replicating ARSs.

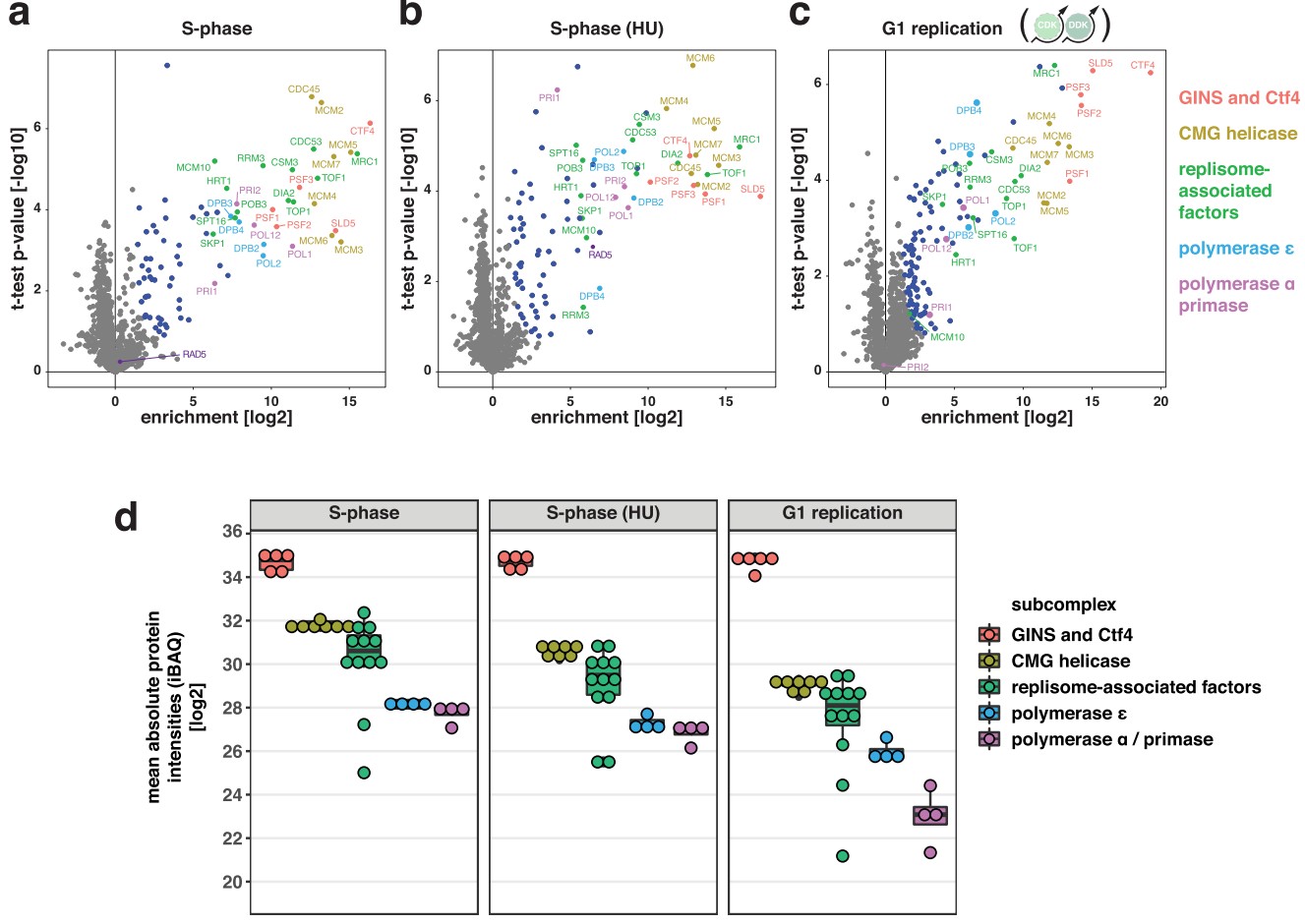

**Fig. 2 | G1 replisomes differ quantitatively in subunit composition from S-phase replisomes. a–c** Replisomes in S- and G1-phase contain the same set of proteins. Cells were synchronously (**a**) released into S-phase, (**b**) arrested in S-phase using hydroxyurea (HU), or (**c**) G1 replication was induced for 3 h using the CDK/DDK bypass system. Replisomes were affinity-purified via GFP-tagged GINS-subunit Psf2 and analyzed by mass spectrometry. Volcano-plots show the enrichment of proteins in GFP-tagged samples versus untagged control samples. Colors indicate statistically significantly enriched proteins and replisome sub-complexes as indicated. Data from n = 3 biological replicates. **d** G1 replication induces fewer replisomes compared to S replication, and additionally polymerase α/primase association is reduced. Label-free quantification and comparison of the datasets shown in (**a–c**) using intensity-based absolute quantification (iBAQ). Boxes indicate the median with the first and third quartile of the sub-complexes, whiskers indicate the minimum and maximum (calculated by extending the box by 1.5 inter-quartile range). Mean iBAQ values for individual proteins are plotted as circles. Data from n = 3 biological replicates.

## G1 replisomes differ from S replisomes

We assessed whether replisome composition differs in G1-phase versus S-phase. To purify replisomes, we immunoprecipitated the GINS complex[52], an integral part of the replicative CMG helicase (Cdc45–Mcm2–7–GINS), via a GFP-tag on its Psf2 subunit and measured replisome composition by label-free quantitative mass spectrometry (qMS). As a benchmark, we purified replisomes from untreated S-phase cells or HU-treated S-phase cells and compared them to untagged control strains. Replisomes purified from these conditions had a protein composition consistent with previous studies (Fig. 2a, b)[52]. The abundance of individual replisome sub-complexes in the final purification varied substantially (Fig. 2d), allowing us to identify sub-complexes that either interact transiently during S-phase or dissociate from replisomes at different rates during purification (Supplementary Fig. 2a, Supplementary Table 3). When we compared replisomes from S-phase and HU-treated cells, we observed a twofold reduced abundance of CMG helicases in the HU sample (Fig. 2d), but additional association of the DNA repair protein Rad5 (Fig. 2a, b) which is known to act in response to replication fork stalling in HU-treated cells[53,54].

S-phase and G1 replication replisomes had qualitatively similar protein compositions (Fig. 2a, c), however the G1 sample had an eightfold reduction in assembled CMG, indicating the presence of fewer replisomes and therefore less efficient replication initiation. The leading strand DNA polymerase ε, fork protection complex Mrc1-Tof1-Csm3, topoisomerase Top1, helicase Rrm3, histone chaperone FACT, and ubiquitin ligase SCF-Dia2 all bound in similar relative ratios to both G1 and S replisomes (Fig. 2c, d). In contrast, the association of DNA polymerase α/primase and helicase activator Mcm10 with replisomes was reduced during G1 replication (Fig. 2c, d), suggesting the existence of another layer of cell cycle regulation acting at the step of replication priming and helicase activation.

In the absence of data on replication fork speeds, the qMS data (Fig. 2) in combination with the data on the increase in DNA content per hour (Fig. 1 and Fig. 3) allows for an estimation of replication fork speeds during G1 replication. Under the assumption that 320 replication forks form during S phase, our quantitative proteomics approach suggested 31–58 replication forks during G1 replication. Accordingly, while S-phase replication forks would have replication speeds of 1.5 Kb/min, G1 replication forks are estimated to progress at 0.8–1.8 Kb/min during G1 replication (see Methods). Taken together, our qMS data suggest that G1 replisomes form less efficiently but have the same protein composition as S phase replisomes, even though specific factors showed reduced association.

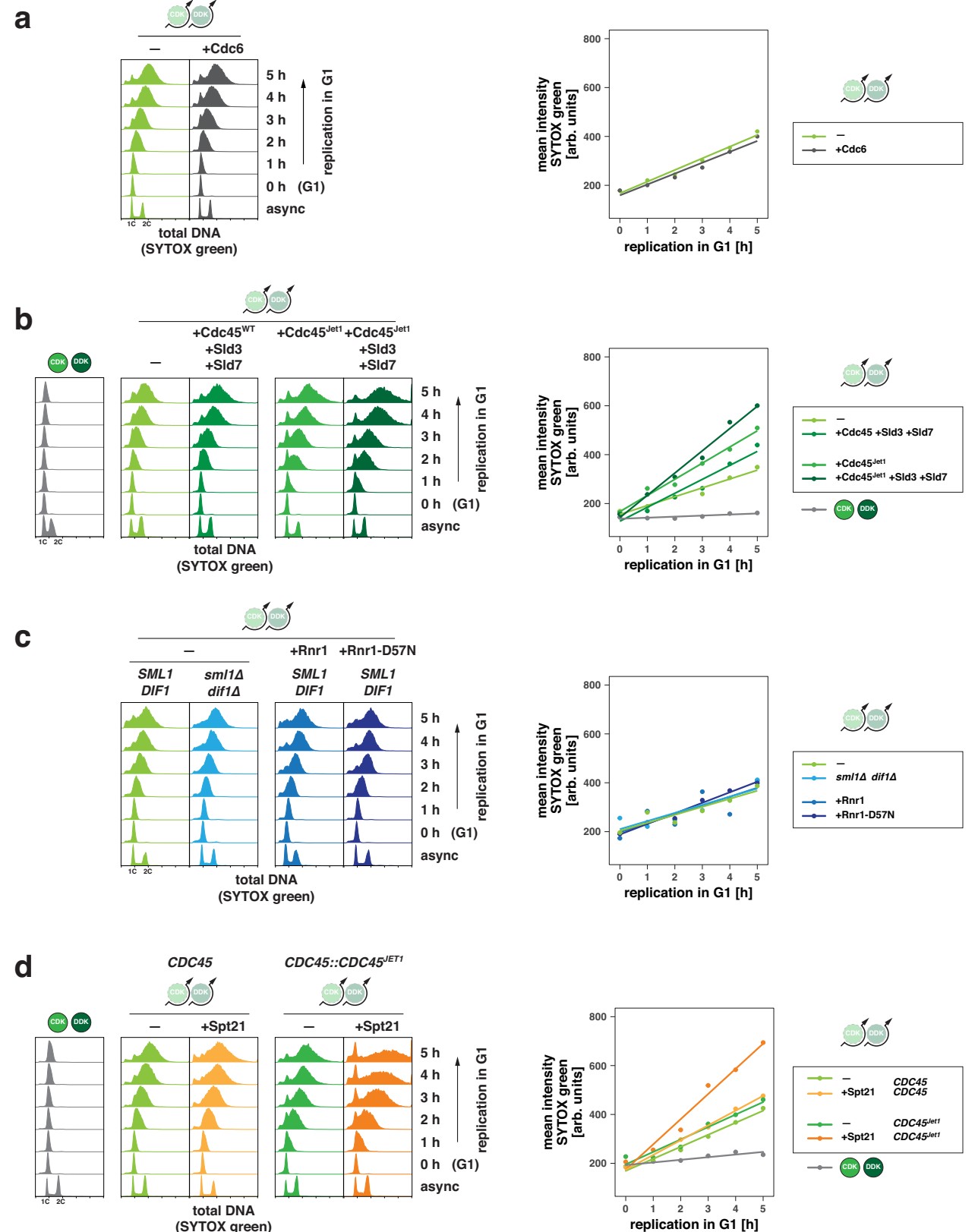

## Factors limiting unscheduled G1 replication

To determine what constrains the formation of replisomes in G1 compared to S-phase we used our flow cytometry-based experimental setup. We considered the following factors could potentially restrict unscheduled replication in G1: depletion of licensed origins, ineffective bypass of CDK and/or DDK phosphorylation, low-abundance of firing factors in G1[55,56], and insufficient supply of dNTPs as well as histones[57–60].

To ask whether licensed origins may become depleted during G1 replication, we further increased origin licensing activity by over-expressing the helicase loading factor Cdc6 which is tightly regulated through degradation at various cell cycle stages[61]. Cdc6

**Fig. 3 | Factors limiting unscheduled G1 replication include availability of firing factors and histones. a** Origin licensing does not limit levels of unscheduled G1 replication. Cells were kept arrested in G1 and induced to replicate DNA using the CDK/DDK-bypass system with or without additional galactose-inducible expression of licensing factor Cdc6. (left) SYTOX green-stained total DNA after induction of G1 replication measured by flow cytometry at indicated timepoints. (right) Quantification of total DNA data in left panel by approximation of a bimodal distribution and calculating means of individual normal distributions. The average mean from 5 fits per timepoint is shown together with a linear regression. Data are representative of $n = 2$ biological replicates. **b** CDK/DDK bypass for unscheduled replication in G1 is limited by the availability of initiation factors and efficient bypass of CDK-control. Experiment and analysis as in (**a**) but cells additionally expressed high levels of firing factors Sld3/Sld7 as well as helicase component Cdc45 either as a wild-type or as a *JET1*-mutant, which bypasses CDK-regulation of Sld3. Data are representative of $n = 2$ biological replicates. **c** Activation of ribonucleotide reductase (RNR) does not lead to an increase of G1 replication. Experiment and analysis as in (**a**) but cells additionally expressed the indicated *RNR1* alleles or lacked negative RNR regulators Sml1 and Dif1. Data are representative of $n = 2$ biological replicates. **d** Increasing histone availability through transcription factor Spt21 increases unscheduled replication in G1. Experiment and analysis as in (**a**) but cells were additionally expressing high-levels of transcription factor Spt21 which regulates transcription of histone genes and is normally degraded in G1. Data are representative of $n = 2$ biological replicates.

overexpression did not affect the total amount of replication in G1 nor its initiation kinetics (Fig. 3a), indicating that the number of licensed origins in G1 cells does not limit replication in G1 in the CDK/DDK bypass setup.

Origin firing is known to be limited by the availability of replication initiation proteins[55,56]. Therefore, in addition to Dbf4, Dpb11, and Sld2-T84D we also expressed Sld3, Sld7, and Cdc45 from a galactose-inducible promoter and observed a moderate increase in DNA synthesis compared to the original CDK/DDK bypass strain (Fig. 3b). This suggests unscheduled G1 replication is constrained by the low abundance of firing factors.

The $CDC45^{JET1}$ allele is suggested to enhance binding of the Cdc45 protein to Sld3 and thereby bypass the requirement for CDK-phosphorylation of Sld3[37,38]. Cdc45[Jet1] expression led to increased G1 replication, detectable after 1 h of induction when combined with the basic CDK/DDK bypass system (Fig. 3b). In contrast, deleting the DDK-antagonizing PP1-phosphatase targeting subunit *RIF1*[62–64] or expressing an overactive, degradation-resistant allele of *DBF4* ($dbf4^{RxxL-4A}$)[48–50] did not alter the extent or the kinetics of unscheduled replication in G1 suggesting that DDK activity is not limiting in our setup (Supplementary Fig. 3a). Taken together, further facilitating the bypass of CDK control of origin firing increases G1 replication, likely due to enhanced replication initiation.

Given levels of dNTPs and histone proteins rise at the G1-S transition to ensure effective genome replication[57,59], we tested whether availability of dNTPs and histones limits G1 replication. To increase dNTP concentrations in G1, we either deleted the ribonucleotide reductase inhibitors *SML1* and *DIF1* and/or over-expressed the catalytic subunit *RNR1* of ribonucleotide reductase as a wild-type or a D57N-allele, which is insensitive to feedback inhibition[65–68]. Enhanced DNA synthesis was not observed in any of these conditions, suggesting that either concentrations of dNTPs are not a bottleneck to unscheduled replication in G1 or additional G1-specific mechanisms exist, which suppress the rise of dNTP levels and affected our ability to experimentally induce dNTP synthesis in G1 (Fig. 3c).

Cell cycle regulation of the histone synthesis-promoting transcription factor Spt21 restricts expression of core histones to S-phase[69]. Indeed, transcript levels of all canonical histone genes were low in G1 and G1-replicating cells but strongly induced in S-phase (Supplementary Fig. 3b). Therefore, we tested whether ectopic expression of SPT21 may facilitate G1 replication. Indeed, high levels of Spt21 resulted in a marked increase in G1 replication induced by the CDK/DDK bypass system (Fig. 3d), indicating that lack of histone synthesis constitutes a bottleneck for unscheduled replication in G1. Moreover, expression of SPT21 increased DNA synthesis synergistically with the $CDC45^{JET1}$ allele (Fig. 3d). Thus, efficient DNA replication can be reconstituted in G1 cells with major bottlenecks being an effective bypass of CDK control of origin firing and the low availability of histones in G1. These two factors could have complementary effects: While Cdc45[Jet1] enhanced the efficiency of replication initiation as judged by the early increase of DNA content already 1 h after induction, Spt21 may improve replication elongation (note the estimation of fork speeds described above) by promoting more efficient histone synthesis.

## Consecutive G1- and S-phase replication generates DNA damage

To understand the consequences of unscheduled replication in G1 and how and when loss of replication control is detected, we induced unscheduled G1 replication and then released cells into the cell cycle. After G1 replication, cells entered and progressed through S-phase similar to control cells but subsequently entered cell cycle arrest suggesting DNA damage had occurred (Supplementary Fig. 4a). Measurement of phosphorylated H2A (γH2A), a DNA damage marker, revealed that G1 replication induced low levels of γH2A in G1 (Supplementary Fig. 4b), however passage through S-phase resulted in substantial accumulation of γH2A in late S/early M (Fig. 4a, b; 40 min after release). The γH2A increase was accompanied by the activation of the DNA damage checkpoint, as evidenced by hyper-phosphorylation of checkpoint kinase Rad53 (Fig. 4b). Checkpoint activation was dependent on the DNA damage checkpoint mediator Rad9, but not the replication checkpoint mediator Mrc1 (Supplementary Fig. 4c, d). The checkpoint was not activated during G1 replication, even though replicating G1 cells were checkpoint-proficient (Supplementary Fig. 4e, f). Thus, G1 replication per se does not trigger checkpoint activation, suggesting no widespread stalling of G1 replication forks.

We hypothesized that the DNA damage checkpoint was not sufficiently sensitive to detect the low levels of damage signal arising from the small numbers of active replisomes operating during G1 replication as determined by mass spectrometry (Fig. 2d). To test this idea, we introduced the *DDC1-RAD9-fusion* allele, which increases the sensitivity of checkpoint signaling[70,71], and observed that Rad53 was activated in response to G1 replication (Fig. 4c, d). These cells showed decreased amounts of DNA synthesis, both during G1 replication and the subsequent release (Fig. 4c), and the γH2A signal was largely suppressed (Fig. 4d). Thus, these cells activate the checkpoint and thereby would inhibit replication at the levels of both new origin firing[72–77] and elongation of ongoing forks[76,77], which limits DNA synthesis and prevents further DNA damage. These data suggest that endogenous checkpoint controls lack sufficient sensitivity to detect unscheduled replication in G1 but that a more sensitive checkpoint could prevent excessive unscheduled replication and the occurrence of DNA damage.

To test the hypothesis that DNA replication in S-phase following G1 replication gave rise to DNA damage, we conditionally depleted the firing factor Sld3 from cells (Fig. 4e) using an optimized auxin-inducible degron system[78–80]. Induction of G1 replication, followed by Sld3 degradation, and release into S-phase allowed us to shut off replication initiation specifically in S-phase, as observed in both control cells and cells that had undergone G1 replication (Fig. 4e). Notably, suppressing replication initiation in S-phase also suppressed the occurrence of DNA damage and the activation of the DNA damage checkpoint (Fig. 4f). This experiment strongly indicated that conflicts between G1 and S replication led to the occurrence of DNA damage. To determine whether over-replication, which involves re-licensing of

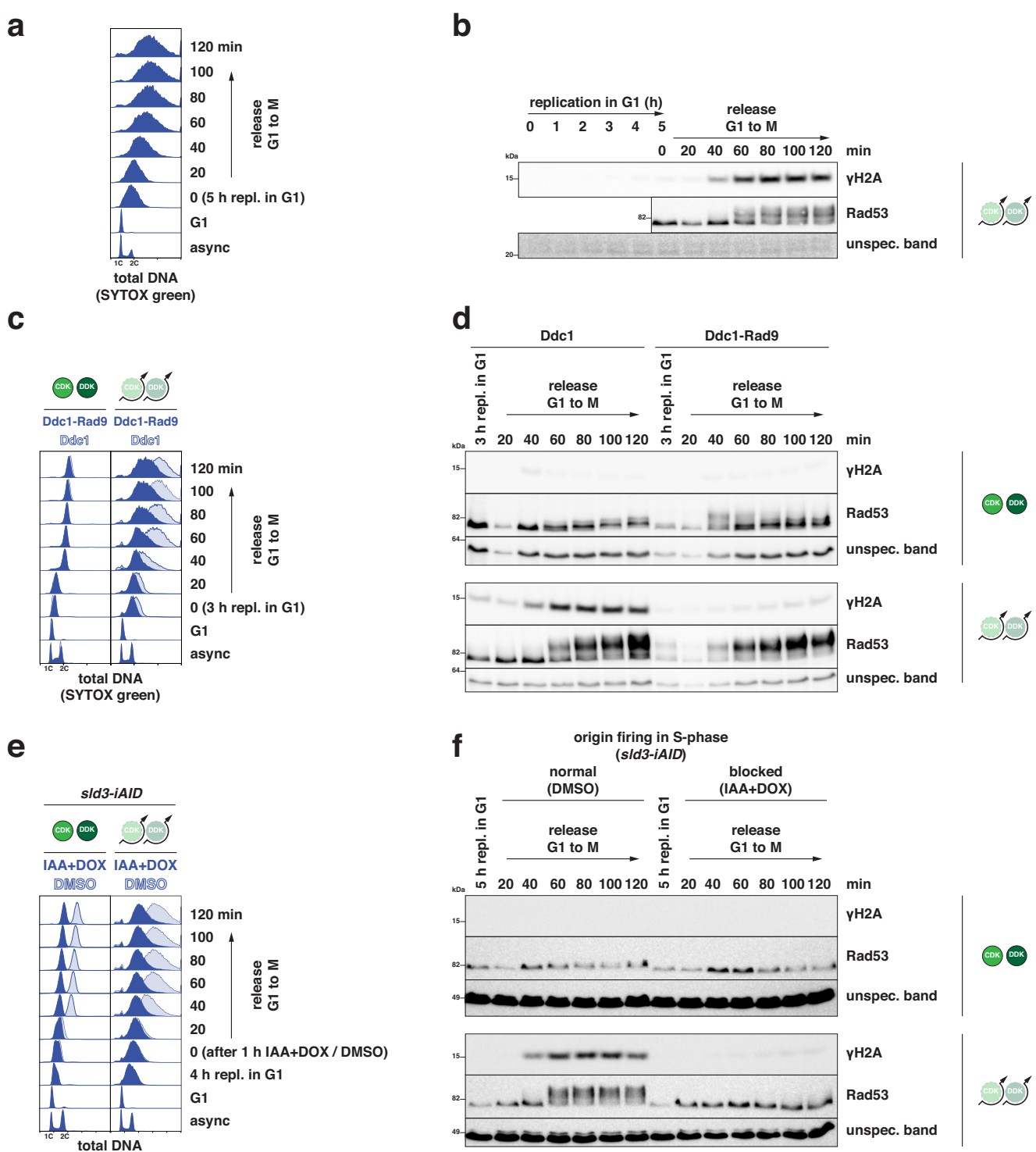

**Fig. 4 | Unscheduled G1 replication induces DNA damage upon S-phase replication. a,b** High levels of DNA damage and checkpoint activation occur late in the subsequent S-phase. To test the consequences of unscheduled G1 replication, cells were released from G1 arrest after 5 h of G1 replication (CDK/DDK bypass) into the cell cycle (M phase arrest, nocodazole) and followed for indicated times. **a** SYTOX green-stained total DNA content as measured by flow cytometry at the indicated timepoints showing S-phase replication at 20–40 min after release. **b** Western blots detecting levels of γH2A and Rad53, for which phosphorylated forms are visible by gel shift, at the indicated timepoints show strong induction of DNA damage from 60 min after release. Data are representative of *n* = 2 biological replicates. **c**, **d** A hyper-sensitized DNA damage checkpoint restricts G1 replication and prevents DNA damage induction after release. Experiment as in (**a**, **b**) with strains expressing a Ddc1-Rad9 fusion protein as a second copy of Ddc1, which leads to hyper-sensitized checkpoint controls also during G1 replication. **c** Flow cytometry data as in (**a**); **d** western blots as in (**b**). Data are representative of *n* = 2 biological replicates. **e**, **f** S-phase replication is required for DNA damage induction and checkpoint activation. Experiment as in (**a**, **b**) with strains harboring a *sld3*-iAID degron allele to conditionally induce Sld3 degradation and suppress further origin firing during S-phase release. Depletion of Sld3 was triggered by addition of 3 mM auxin (IAA) and 20 μg/ml doxycycline (DOX) in the last hour of G1 replication before release and suppressed DNA damage accumulation in and after S-phase. **e** Flow cytometry data as in (**a**); **f** western blots as in (**b**). Data are representative of *n* = 2 biological replicates. Source data are provided as a Source Data file.

origins, caused the observed DNA damage also during G1 replication, we depleted the licensing factor Cdc6 during G1 replication, using a similar degron approach (Supplementary Fig. 4g, h). Cdc6-depleted cells undergoing G1 replication synthesized less DNA (Supplementary Fig. 4g) and had substantially reduced levels of DNA damage, as indicated by γH2A (Supplementary Fig. 4h).

In addition, we also investigated whether G1 replication per se impaired replication initiation in the subsequent S-phase. To ascertain origins fired normally, we measured DNA synthesis in early S-phase through EdU-seq of HU-treated cells. EdU-seq signals in S-phase following G1 replication were found at early firing origins and were similar to those observed in HU-treated controls (Supplementary Fig. 5a, b). We also tested whether composition of S-phase replisomes would change due to prior G1 replication. Purifying replisomes and quantifying their composition as described above, we observed that S-phase replisomes showed a highly similar protein composition, independently of whether G1 replication was induced or not (Supplementary Fig. 5c–e). Only for the replisome-associated protein Mcm10 we observed IBAQ values that were fourfold lower after G1 replication compared to a normal S-phase (Supplementary Fig. 5e).

Together, these data demonstrate that G1 replication does not affect the initiation of replication and the formation of replisomes in the subsequent S-phase. Successive rounds of replication initiation in G1 and S-phase, and to a lesser extent already during G1 replication, promote over-replication which in turn will generate DNA damage.

### Successive G1 and S replication generate single-ended DSBs

To visualize when and where successive G1 and S replication induced DNA damage, we analyzed chromosomes from a time course experiment using pulsed-field agarose gel electrophoresis (PFGE). Full-length chromosomes enter the PFGE gel, while the presence of replication forks or repair structures traps affected chromosomes in the loading slot. Using Southern blot probes against a marker locus (*TRP1*) present on chromosomes 4 and 7 in the analyzed strains, we observed that replication structures were only present during S-phase in control cells (20 min, Fig. 5a, Supplementary Fig. 6a). In contrast, chromosomes were largely retained in the loading slots if cells had previously undergone replication in G1 (Fig. 5a, Supplementary Fig. 6a). The level of retention correlated with the amount of replication induced in G1, when comparing CDK bypass with CDK/DDK bypass conditions (Fig. 5a, Supplementary Fig. 6a). We detected additional signals (smears) below the chromosome bands after 80 min of release, indicative of DNA double-strand breaks (DSBs) and demonstrating that DSBs occur at late timepoints after successive G1 and S replication.

DSBs will be either single-ended or double-ended. To distinguish between these two possibilities, and assuming that DSBs become resected, we used a strand-specific ChIP-seq approach directed against RPA to study the occurence of single-stranded DNA[81]. We chose chromosome 4 as a representative of all chromosomes and observed over-representation of regions around the centromere in total DNA after 3 h of G1 replication and subsequent release when comparing to a control strain, indicating preferential replication of these regions (Supplementary Fig. 6b), consistent with our EdU-sequencing data (Fig. 1c). Regions of single-stranded DNA, as marked by increased RPA binding, appeared only after release from G1 arrest (Supplementary Fig. 6c). RPA preferentially bound to the forward-strand DNA on the right arm of the chromosome and to the reverse-strand DNA on the left arm of the chromosome (Supplementary Fig. 6d). Both short and long chromosomes were affected, as shown by RPA read asymmetry scores normalized for chromosome length (Supplementary Fig. 6e). This asymmetric binding pattern was independent of *RAD52* (Supplementary Fig. 6c–e), indicating that it does not involve recombination processes such as break-induced replication (BIR)[82–84].

To analyze the appearance of the ssDNA-RPA signals, we carried out a time course experiment involving G1 replication induction for 3 h

and then samples being taken at 0, 30, 60, 120, and 180 min following release into S-phase. Using the CDK/DDK bypass conditions, we observed an over-representation of regions around the centromere in total DNA samples after G1 replication on chromosome 4 (representative of all chromosomes), demonstrating again that G1 replication is biased to centers of chromosomes (Fig. 5b). With cells in S-phase (30 min after release), we observed an over-representation of sequences close to origins of replication in total DNA (Fig. 5b) and an RPA pattern (Fig. 5c–e) consistent with single-stranded lagging strand template DNA (Supplementary Fig. 6f, g). Averaging RPA-ChIP-seq data over all yeast chromosomes, we did not detect any chromosome-wide strand-biased RPA binding in control cells after 3 h of G1 replication nor in the first 30 min of the following S-phase (Fig. 5d top).

However, 60 min after release and at later time points, a strand-biased RPA-ssDNA binding pattern developed over entire chromosomes (Fig. 5c, d middle/bottom). G1 replication by CDK bypass compared to G1 replication by CDK/DDK bypass led to RPA signals in the same chromosomal regions after the following S-phase, but with more pronounced strand bias throughout the genome (Fig. 5f, g, Supplementary Fig. 6h). We reasoned that such chromosome-wide RPA-ssDNA strand bias would be generated if single-ended DSBs occurred with a biased orientation and were resected thereafter. In our experiments, G1 replication preferentially initiated around chromosomes centers (Fig. 1c) and traveled toward chromosome ends, recapitulating the inherent distribution of early- and late-replicating origins along chromosomes (Supplementary Fig. 6i). Further replication initiation in S-phase then caused replication collisions that gave rise to single-ended DSBs accompanied by exposure of RPA-covered ssDNA on the forward or reverse strand depending on which direction the colliding replication forks were moving. Focusing our analysis on origins which are active during G1 replication (Fig. 5e), our data supports a model (see below) where stochastic head-to-tail replication collisions between G1 replication forks and tailgating S-phase forks occur with directional bias toward chromosome ends and generate the observed RPA binding-patterns.

G1 replication of a given chromosome will often be incomplete with unterminated replication structures being obstacles of subsequent S replication. Head-to-tail replication collisions therefore are an intrinsic and detrimental consequence of over-replication (see below). Consistently, after induction of G1 replication, only few cells were able to form viable colonies (Fig. 5h). Using whole-genome sequencing, we found that the majority of survivors showed a whole-chromosome aneuploidy of at least one chromosome (Fig. 5i). A possible way how such chromosomal duplications could arise is by complete replication of the affected chromosome during the initial G1 replication. Notably, this would also clear the affected chromosome of replication structures and avoid later replication collisions. These data therefore suggest that surviving unscheduled replication initiation could involve complete over-replication of entire chromosomes during G1 replication or subsequent repair.

### Sporadic G1 replication generates replication collisions and genome instability

Since the increase in RPA-bound ssDNA did not scale with the amount of unscheduled G1 replication, we asked whether a single or few sporadic events of unscheduled replication initiation per cell could trigger similar cellular responses and genome instability. To test this idea, we devised an experimental setup to trigger sporadic, unscheduled replication in G1 by enhancing the physical interaction between firing factors Dpb11 and Sld2. We fused various split-Venus tags with Dpb11 and Sld2 proteins, which were expressed at similar levels (Supplementary Fig. 7a) but stabilized the interaction to different degrees (Fig. 6a, Supplementary Fig. 7b). The combination of Dpb11-VN and VC-Sld2 yielded the highest Venus fluorescence intensity indicating that it stabilized the physical interaction most effectively (Fig. 6a,

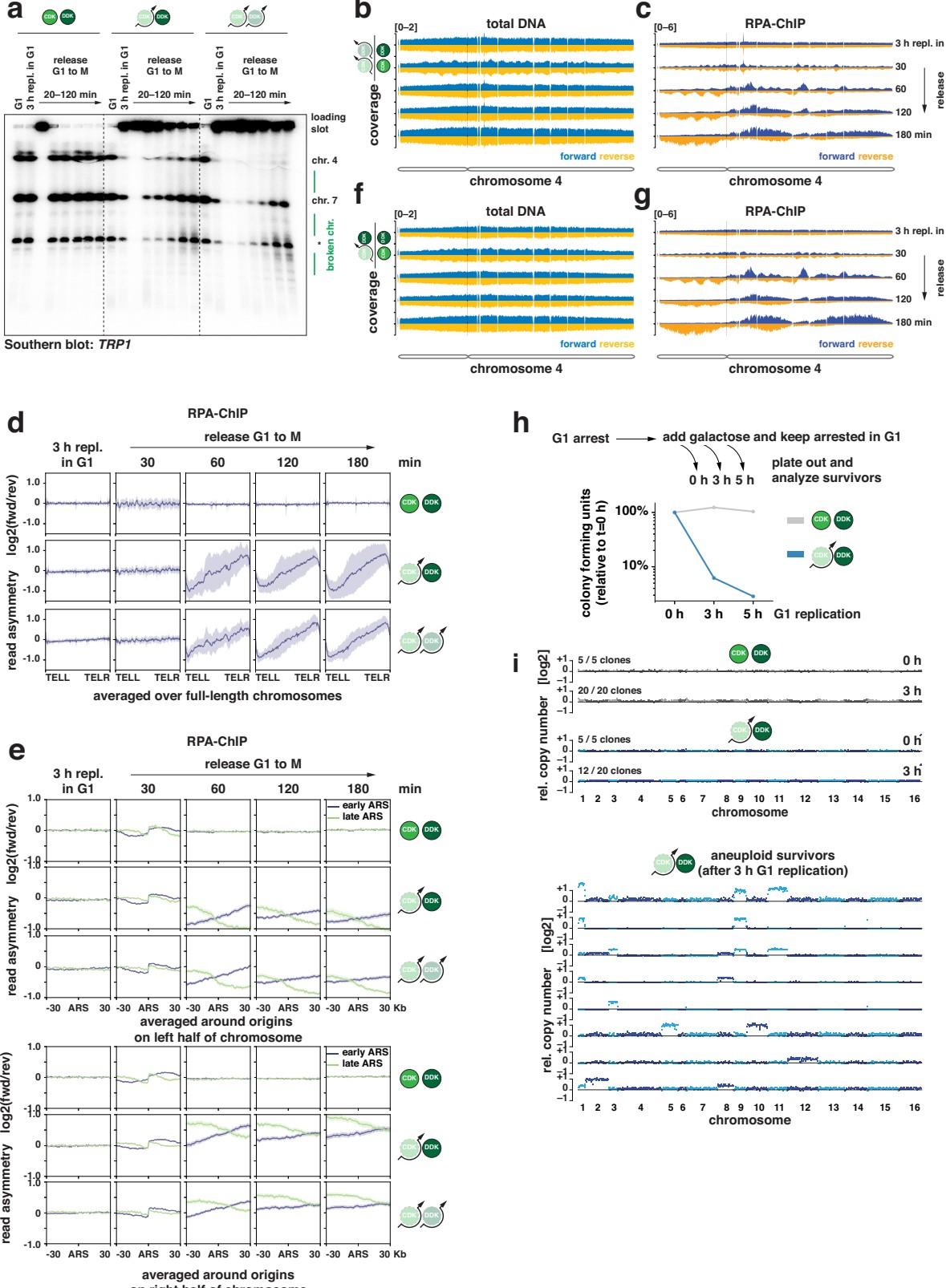

Supplementary Fig. 7b). We assessed the extent of sporadic G1 repli-
cation by arresting cells in G1 in the presence of EdU and measured
DNA synthesis at various times. We found that DNA replication initi-
ated only in a sub-population of cells and relatively little DNA was
replicated in these cells compared to the inducible G1 replication
systems used before (Supplementary Fig. 7c; compare to Fig. 1b). The

combination of Dpb11-VN and VC-Sld2 yielded the highest level of
replication in G1, consistent with the interaction data (Fig. 6a, Sup-
plementary Fig. 7b). Thus, sporadic initiation of DNA replication in G1
can be mediated by a synthetic Venus-bridged Dpb11-Sld2 complex.

To test if this sporadic system recapitulates the hallmarks of
unscheduled G1 replication, we arrested cells in G1 and subsequently

**Fig. 5 | Successive G1 and S replication generate single-ended DSBs from head-to-tail fork collisions, resulting in an asymmetric pattern of RPA-bound ssDNA on chromosome arms. a** Chromosome breaks occur after release from unscheduled G1 replication. Replication was induced in G1 and cells were afterwards released into S-phase in the presence of nocodazole as in Fig. 4. Samples were taken at the indicated timepoints and chromosomes were separated by pulsed-field gel electrophoresis. A probe directed against the *TRP1* gene was used to visualize endogenous and ectopic loci on chromosomes 4 and 7. Low molecular weight fragments indicate chromosome breakage, the asterisk indicates an unspecific band. Data are representative of *n* = 2 biological replicates. **b,c** RPA accumulates on chromosomes with strand bias. Representative traces of total DNA and RPA-ChIP from chromosome 4. G1 replication was induced by CDK/DDK bypass and cells were released to nocodazole-containing medium for the indicated times. After cross-linking with formaldehyde, RPA-bound DNA was purified and sequenced. Reads were separated by strand and the coverage traces of both strands were compared between CDK/DDK-bypass and control strains. Reads mapping to the forward strand are shown in light blue/dark blue; reads mapping to the reverse strand are shown in yellow/orange. The scale of the *y*-axis is given at the top-left corner of each panel. The dashed line indicates the position of the centromere. Data are representative of *n* = 2 biological replicates. **d,e** RPA accumulates on chromosomes in a characteristic, asymmetric pattern with strand bias to left and right halves of the chromosomes. Data from the experiment in (**b/c**) was analyzed by averaging the read asymmetry (log2-ratio of depth-normalized RPA-ChIP-seq reads mapping to forward and reverse strands) at different genomic features. **d** Average read asymmetry (dark) ± SD (light) over full-length chromosomes reveals an opposite strand bias for both chromosome halves. **e** Average read asymmetry in 60 Kb windows around early-(blue) or late-firing origins of replication (ARS) separated by left/right halves of the chromosome show that strand bias is strongest upstream of early-firing origins and downstream of late-firing origins for the left half of chromosomes and vice versa for the right half of chromosomes. Data are representative of *n* = 2 biological replicates. **f, g** same experiment as in (**b**)/(**c**) but G1 replication was induced by CDK bypass. **h** G1 replication is highly toxic. Replication was induced in G1-arrested cells by addition of galactose (CDK bypass) and cells were plated on non-selective agar plates at the indicated timepoints after induction of G1 replication. Resulting colonies were counted after incubation for 2 days at 30 °C. Mean colony forming units (relative to t = 0 h) at the indicated times after induction of unscheduled G1 replication are plotted on a logarithmic scale for wild-type control cells (gray) and CDK bypass cells (blue). Data are representative of *n* = 2 biological replicates. **i** Survivors of unscheduled G1 replication acquire aneuploidies. Colonies grown from single clones after 0 h and 3 h of G1 replication (experiment in (**h**)) were analyzed by whole genome sequencing. The relative copy number of 1 Kb bins is plotted in logarithmic scale (alternating colors indicate different chromosomes). Clones without obvious copy number variations were grouped as indicated by the numbers in the top left corners. (top) No copy number variations were detected before G1 replication and in only 60% of survivors that had undergone 3 h of G1 replication. (bottom) Survivors of G1 replication that acquired copy number variations. Note that only full chromosome aneuploidies were observed. Data are representative of *n* = 2 biological replicates. Source data are provided as a Source Data file.

followed them through one round of the cell cycle until the next G1-phase (Fig. 6b, Supplementary Fig. 7d). We detected phosphorylated Rad53 at 50–60 min after release (Fig. 6c), the level correlating with the amount of replication in G1, as shown by comparison of the most effective strain expressing VC-Sld2 to the less effective Sld2-VC strain (Fig. 6b, c). Consistent with this, G1 replication triggered by the sporadic system also resulted in cell cycle arrest (Fig. 6b, Supplementary Fig. 7d) similar to strains after G1 replication by CDK/DDK bypass (Supplementary Fig. 4a) with the number of arrested cells being proportional to the amount of G1 replication across different strains (Supplementary Fig. 7c). Thus, sporadic replication in G1 occurs in a sub-population of cells that contains high levels of Venus-bridged Dpb11-Sld2 and results in checkpoint activation and cell cycle arrest after S-phase.

To assess if the sporadic system also leads to strand-biased detection of RPA-bound ssDNA on chromosome arms, we conducted a strand-specific RPA-ChIP-seq experiment where we arrested cells for 5 h in G1 and then released them for 2 h to M phase. While we did not observe strand-biased RPA binding in control cells expressing an interaction-deficient *VC-sld2-T84A* allele, we found that RPA-bound preferentially to the forward strand on the right arm of chromosome 4 and to the reverse strand on its left arm (Fig. 6d, e). Such asymmetry was only detected on long yeast chromosomes that also contain many origins (Supplementary Fig. 7e, f), suggesting that origin-rich chromosomes are more likely to engage in replication in G1 induced by the sporadic system and that G1 replication is a rare event in this system. This notion is consistent with the limited amount of DNA synthesis measured during G1 (Supplementary Fig. 7c). Thus, strand-biased RPA binding on chromosome arms can be observed under conditions where only one or few origins initiate in an unscheduled manner during G1.

To determine if and how different levels of sporadic induction of unscheduled G1 replication cause genome instability, we selected strains expressing VC-tagged Sld2 (*VC-SLD2* and *SLD2-VC*) as they showed different levels of Dpb11-Sld2-complex formation and replication in G1 (Fig. 6a, Supplementary Fig. 7c). Cultures of these strains were grown from single cells to saturation to determine gross chromosomal rearrangement (GCR) rates using an established assay[85]. GCRs are potent drivers of genome instability and frequently observed in cancer cells. We measured a highly increased GCR rate for the *SLD2-VC* (-1000-fold compared to control) strain and an even higher GCR rate for the *VC-SLD2* (-5000-fold) strain (Fig. 6f, Supplementary Fig. 7g), suggesting that levels of genome instability correlated with the amount of sporadic G1 replication. Furthermore, cultures with increased levels of sporadic replication in G1 (*VC-SLD2*) showed decreased viability on non-selective medium, whereas cultures with lower levels of sporadic replication in G1 (*SLD2-VC*) had normal viability (Fig. 6f). Thus, our data suggest that unscheduled G1 replication induces genome instability and cell death, even when only single or few replication origins per cell are affected.

## Discussion

Over-replication has been linked to early stages of carcinogenesis[86] but whether it is a cancer driver remains to be determined. Previous studies in yeast and human cell lines have focused on mis-regulation of helicase loading factors and the induction of over-replication after S-phase[11,86]. In contrast, many oncogenes act by de-regulating the G1-S transition, raising the potential of unscheduled DNA replication in G1 or early S-phase. Here, we induced unscheduled G1 replication in engineered budding yeast systems to reveal details of the molecular mechanism and cellular consequences of this toxic process.

We found that unscheduled helicase activation in G1 induced replication from canonical origins on all chromosomes and re-initiation at single origins was a rare event during G1 (Fig. 1c). Early-replicating origins were prone to both G1 replication and over-replication in our assays (Fig. 1d). The non-random distribution of over-replicated DNA was also observed in a recent study, implying that specific origins tend to participate in over-replication[87]. Such a preference has also been observed in cancer cells exposed to an experimental therapeutic strategy that induces overt over-replication[88]. Over-replication induced by unscheduled helicase activation in G1 appears to differ in this regard from over-replication induced by unscheduled helicase loading in M phase, which re-initiates from a different set of replication origins that are flanked by specific re-initiation promoting sequence elements[89]. This difference not only shows that regulated helicase loading is crucial for the establishment of the replication program, it also highlights that we cannot easily extrapolate findings from previous systems that induce over-replication after S-phase to unscheduled DNA replication in G1.

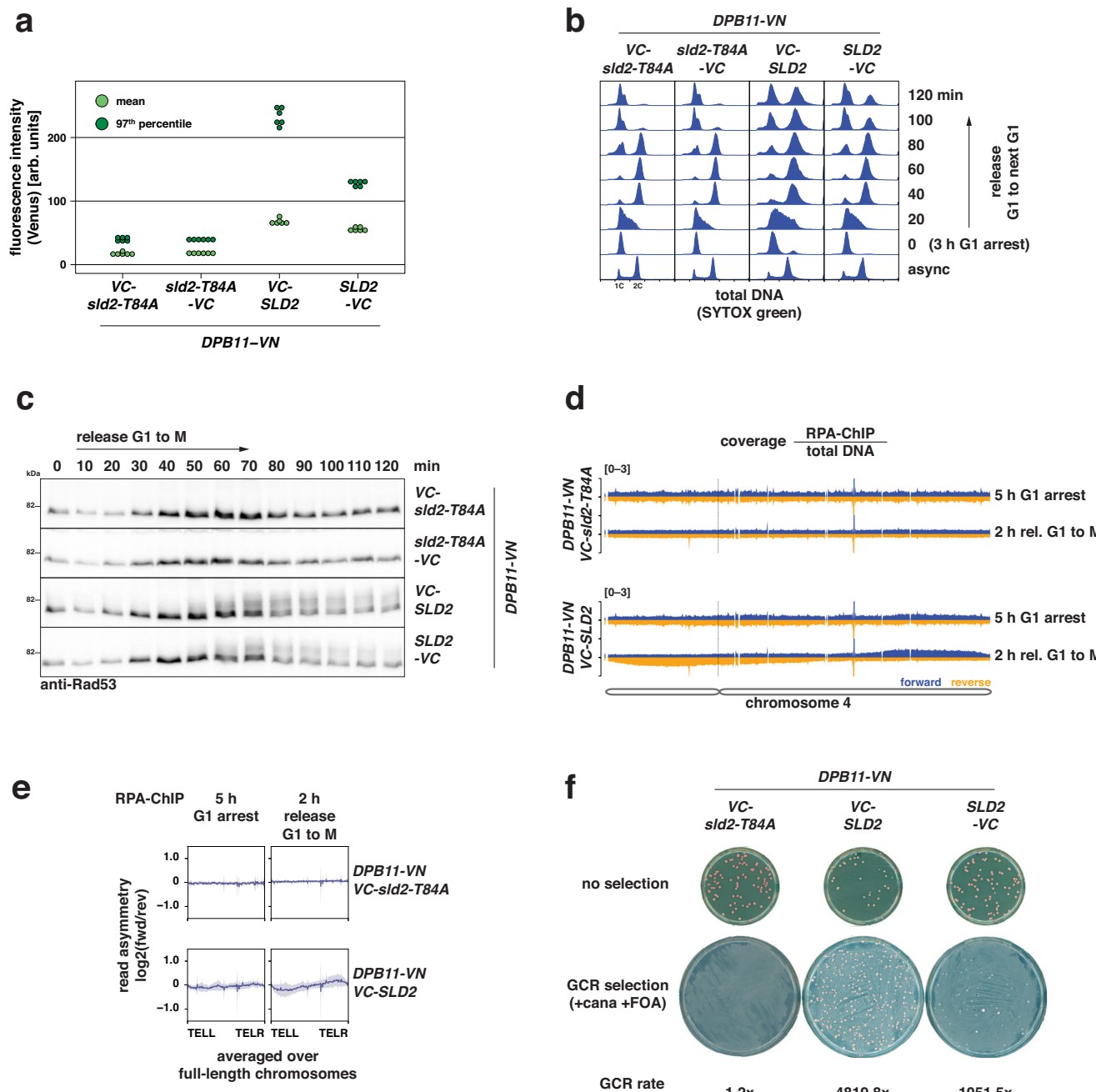

When compared to replication in S-phase, bulk replication in G1 progressed approximately tenfold slower. As similar observations were made for systems of unscheduled replication in M phase[11], we reasoned that additional factors may be constraining replication outside S-phase. Our study lacks DNA combing data, which would allow to determine rates of replication elongation and initiation. Nonetheless, our data suggest that both inefficient replication initiation and inefficient replication elongation may contribute to overall slow replication (Fig. 3). For example, we show that promoting histone synthesis in G1, which is normally a key feature of S-phase[59] and affects the rate of replication elongation[90,91] accelerated G1 replication by approximately twofold (Fig. 3d), suggesting that histone protein availability is a major bottleneck to replication in G1, even though repression of histone synthesis has only minor effects on S-phase length in budding yeast[92]. Our mass spectrometry-based quantification also revealed a reduced association of DNA polymerase α/primase with G1 replisomes (Fig. 2d).

Polymerase α/primase has been proposed to be cell cycle-regulated[93,94] and was found to be phosphorylated by CDK[95]. Such phosphorylation could regulate its association with the replisome and, indeed, the efficiency of replication initiation in S-phase is decreased if protein levels of polymerase α fall below a threshold[96]. It is unclear whether cell cycle control of polymerase α/primase would influence primarily replication initiation or elongation or both, but nonetheless we demonstrate that studying unscheduled replication in G1 facilitates the identification of new targets of cell cycle control.

Our study addresses the question of how cells respond to unscheduled replication and how G1 replication induces DNA damage. We show that G1 replication compromises genome stability and it does so specifically due to conflicts of G1 replication forks with subsequently initiated S replication forks (Fig. 4e, f). The initial G1 replication is not detected by cellular checkpoint controls, likely because it is carried out only by relatively few replisomes. Consistently, we observe

**Fig. 6 | Low levels of sporadic G1 replication also generate head-to-tail fork collisions and genome instability. a** Split-Venus tags (VN/VC) stabilize the physical interaction between Dpb11 and Sld2. Fluorescence intensity of cells expressing Dpb11-VN and Sld2 tagged at either N- or C-terminus with split-Venus fragment VC. Direct interaction of Dpb11 and Sld2 allows for the formation of a covalent link between VN and VC. The *sld2-T84A* mutation abolishes the interaction with Dpb11 and serves as control. Mean (light green) and 97th percentile (dark green) of split-Venus fluorescence intensity were measured by flow cytometry in log-phase cells. Data from $n = 6$ biological replicates. Additional combinations are shown in Supplementary Fig. 7b. **b** Venus-stabilized interaction of Dpb11 and Sld2 results in cell cycle arrest, but not in *sld2-T84A* controls. Cells expressing the indicated combinations of split-Venus-tagged Dpb11 and Sld2 alleles were first arrested in G1, then released and followed for one cell cycle into the next G1 phase. SYTOX green-stained total DNA content was measured by flow cytometry. **c** Venus-stabilized interaction of Dpb11 and Sld2 causes DNA damage checkpoint activation after S-phase, but not in *sld2-T84A* controls. Experiment as in (**b**). Rad53 and phosphorylated isoforms show checkpoint activation at indicated timepoints after G1 release and were detected by western blot. Data are representative of $n = 2$ biological replicates. **d,e** Venus-stabilized interaction of Dpb11 and Sld2 results in asymmetric, strand-biased RPA binding on chromosome arms, but not in *sld2-T84A controls*.

Cells with a Venus-stabilized Dpb11-Sld2-interaction (*VC-SLD2*) or an interaction-deficient control (*VC-sld2-T84A*) were arrested in G1 for 5 h and subsequently released to nocodazole-containing medium. Cells were cross-linked with formaldehyde at the indicated timepoints and RPA-bound DNA was isolated and sequenced. Reads were separated by strand. **d** RPA-coverage of the forward (dark blue) and reverse (orange) strand of chromosome 4 at the indicated timepoints, normalized by total input DNA. The scale of the *y*-axis is given at the top-left corner of each panel. The dashed line indicates the position of the centromere. **e** Read asymmetry (log2-ratio of RPA-ChIP-seq reads mapping to forward and reverse strand) was averaged over full-length chromosomes at indicated timepoints. Data are mean log2-ratio ± SD from $n = 2$ replicates. **f** High levels of genome instability are caused by Venus-stabilized interaction of Dpb11 and Sld2. Cells with the indicated genotypes were subjected to an assay scoring gross chromosomal rearrangements (GCRs) via loss of a *CAN1::URA3* cassette that was integrated at the endogenous *CAN1* locus (-33 Kb from the end of chromosome 5). Representative control ($10^{-6}$ dilution) and GCR selection plates (undiluted) are shown together with relative GCR rates. Absolute GCR rates are given in Supplementary Fig. 7g and were calculated from $n = 8$ replicates per condition. Source data are provided as a Source Data file.

that S-phase replication commences with normal kinetics after release from the G1 arrest, initiating from the same replication origins and with similar replisomes as during an unperturbed S-phase (Supplementary Fig. 5), but high levels of DNA damage occur during or after this S-phase. We conclude a model (Fig. 7) whereby head-to-tail collisions of DNA replication forks are central to this DNA damage induction. G1 replication will leave behind replication forks (Fig. 7a, b) and initiation of replication in S-phase will generate new replication forks that have the propensity to tailgate onto G1 replication forks (Fig. 7a, c, d), no matter which strand is being replicated. Head-to-tail collisions of G1 and S replication forks will generate single-ended DSBs (Fig. 5a, d), which subsequently could expose single-stranded DNA through DNA end resection consistent with the strand-biased appearance of single-stranded DNA (Fig. 5d). The mechanism of DSB induction through head-to-tail replication collisions is thus similar to what has been proposed for over-replication induced by unscheduled helicase loading[14,97] and blocked replisome progression[34]. These collisions will initially be avoided (but not ultimately prevented) if the G1 replication fork encounters a fork in head-to-head orientation, leading to termination before the tailgating fork arrives. In this case, however, a new problem arises because the tailgating S-phase fork will now remain unterminated because it lacks a termination "partner" and therefore will potentially lead to a head-to-tail collision with a neighboring S-phase fork. Head-to-tail replication collisions can be avoided if the entire chromosome is re-duplicated and replication structures thereby run off chromosome ends (Fig. 7). Consistently, we observed a striking increase in whole-chromosome aneuploidies among survivors of G1 replication (Fig. 5i). These data indicate that re-duplication of an entire chromosome may provide a means to avoid subsequent replication collisions. It also suggests a link between G1 replication and chromosome instability.

The genomic locations of potential replisome collisions are determined by the location of the G1 replication fork relative to its two nearest origins as well as their respective initiation timing. Because helicase activation is itself a stochastic process[98–100], forks will also be resolved stochastically. We observed that G1 replication mimics early S-phase[101] with replication initiating primarily from origins in central regions of chromosomes, including centromeres, but not toward chromosome ends (Fig. 1c). Therefore, G1 replication forks that will be involved in head-to-tail collisions will mainly be moving outwards toward telomeres. It is thus the timing and the relative efficiency of replication origins that shape where over-replication generates single-ended DSBs within the genome.

At this point, we cannot exclude that subtle changes in replisome composition, such as the reduced association of Mcm10, may further

aggravate the problem and hamper the cellular response to replication collisions. We can also not exclude that replication run-off contributes to the occurrence of single-ended DSBs and single-stranded DNA. Such run-off will occur if the product of G1 replication, which is used as the template for S-phase replication, contains DNA nicks (single-strand breaks) or gaps. Indeed, large RPA-coated ssDNA gaps have been observed on the template strand during over-replication in human cells[31]. In contrast, we do not observe evidence for the occurrence of large ssDNA gaps during G1 replication but cannot exclude a contribution of DNA nicks[102]. Taken together, unscheduled G1 replication results in a characteristic signature of single-stranded DNA, which has the potential to serve as a marker for over-replication.

Over-replication from multiple origins may be a rare event given the various endogenous replication control mechanisms. Sporadic unscheduled replication events affecting only single origins and thereby chromosomes may however be more likely, particularly under conditions of deregulated cell cycle control. Our model suggests that even a single event of unscheduled replication will be detrimental to the affected chromosome and can only be resolved if the entire chromosome is re-duplicated. To test this hypothesis, we generated systems to sporadically induce unscheduled replication in G1 (Fig. 6) and observed the same signature of asymmetric, strand-biased RPA accumulation (Fig. 6d). Such asymmetry was detected preferentially on long yeast chromosomes (Supplementary Fig. 6e). We speculate that long chromosomes undergo unscheduled replication more frequently due to the higher number of (early-firing) origins (Supplementary Fig. 6f). We also observed a substantial increase in chromosomal rearrangements (Fig. 6f, Supplementary Fig. S7g), highlighting the potential of rare over-replication events as potent drivers of genome instability.

Several oncogenes deregulate the G1/S transition, trigger a premature S-phase, and cause replication stress in human cell culture systems[103,104]. Mechanistically, replication stress upon oncogene activation was linked to the firing of oncogene-induced origins and replication-transcription conflicts[46,105]. An oncogene-induced premature S-phase has conceptual similarity to the synthetic systems of G1 replication investigated here. We do not find evidence for transcription-replication conflicts occurring during G1 replication in yeast, as judged by the relatively minor occurrence of DNA damage signals in G1. Therefore, future work will be required to compare whether such differences arise from the particularities of how unscheduled replication was induced or from differences between species and chromosome organization. These studies will help reveal the contribution of unscheduled replication to early carcinogenesis.

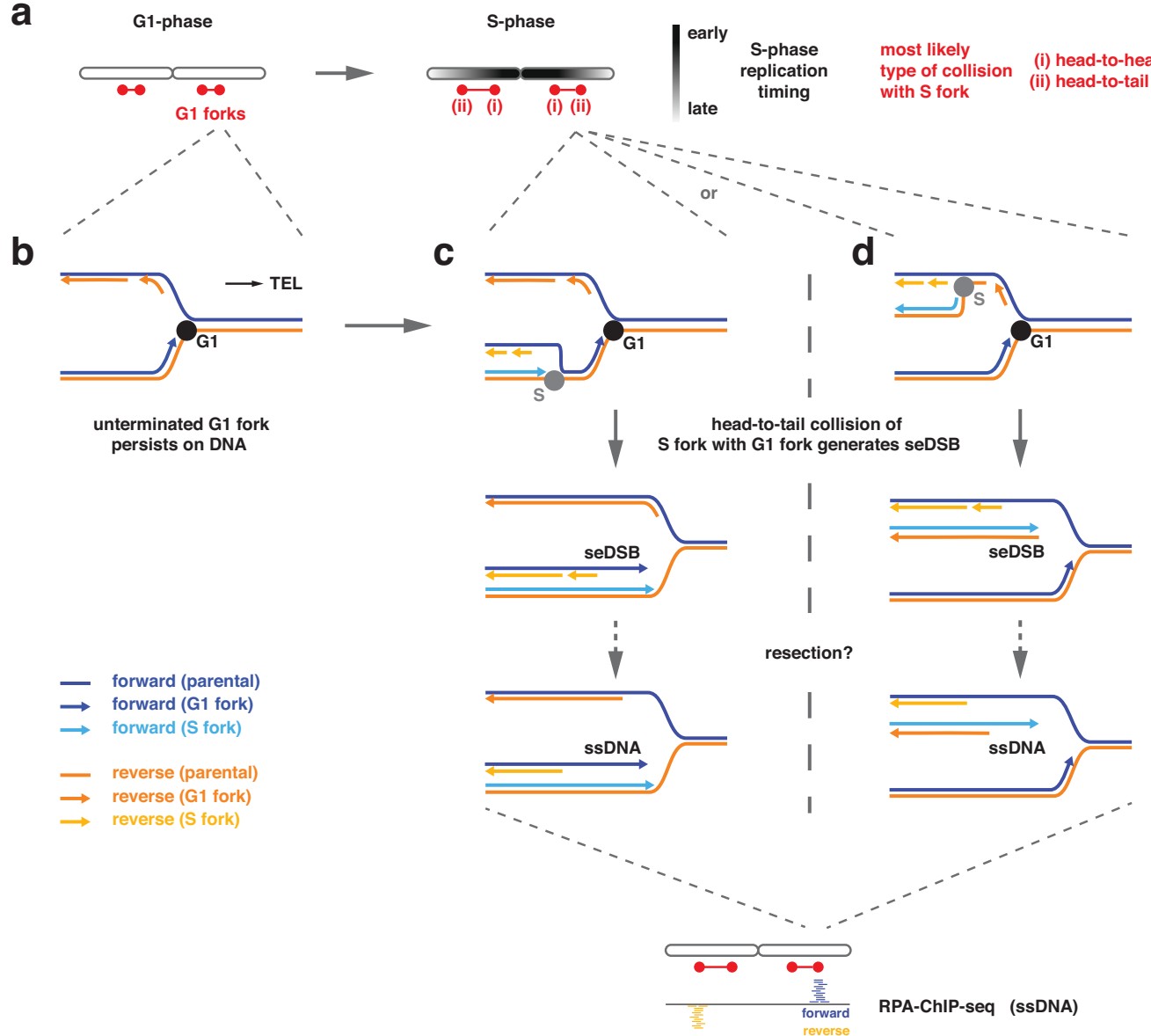

**Fig. 7 | Single-ended double-strand breaks result from head-to-tail collisions of S forks with unterminated G1 forks. a** S-phase replication forks are most likely to collide with G1 replication forks in a head-to-tail-orientation. Several factors determine where these collisions occur. First, G1 replication will usually initiate at early firing origins that are preferentially located close to chromosome centers. G1 replication forks going inward will often be terminated by G1 or S replication forks that also initiate from early firing origins. In contrast, G1 replication forks going outward are less likely to be terminated by a G1 or S-phase replication fork emanating from a late-firing origin reaches and therefore will be more frequently subject of head-to-tail collisions. **b** For simplification, a single replication fork moving to the right telomere is shown. A replication fork moving to the left telomere will encounter the same events just with opposite strand directionality. Unterminated forks persist after unscheduled replication in G1 due to incomplete duplication/missing termination. **c,d** Head-to-tail collision of an S-phase replication fork with an unterminated G1 replication fork. A single-ended double-strand break (seDSB) occurs independent of whether the S-phase replication fork travels on the parental reverse strand (**c**) or the reverse strand synthesized by the G1 replication fork (**d**). Single-stranded DNA (ssDNA) could either be directly exposed during seDSB generation (due to incomplete lagging strand replication) or afterwards by resection. Independent of its generation, the ssDNA will be bound by RPA and gives rise to a strand-biased pattern of reads in RPA-ChIP-seq experiments.

## Methods

### Yeast strains and culture

All yeast strains were constructed in the W303 background using standard methods[106]. Genotypes of all used strains are given in Supplementary Table 1. If not stated otherwise, strains were constructed in the EdU-incorporating background of E3087. Integrative plasmids were linearized prior to transformation and single integration of plasmids was confirmed by PCR. Gene deletions and tags were introduced using a PCR-based protocol.

For cell cycle experiments, cells were grown to log-phase (OD600 of 0.5–0.6) at 30 °C in YP medium supplemented with adenine and either 2% raffinose (inducible G1 replication system) or 2% glucose (sporadic G1 replication system) and synchronized in G1 by adding α-factor (MPIB core facility or GenScript RP01002) to a final concentration of 0.5 μg/ml for *bar1Δ* cells or 10 μg/ml for *BAR1* cells. Additional doses of α-factor were added after each hour of arrest to achieve a stable arrest of *BAR1* cells. Hydroxyurea (Sigma H8627) was added to a final concentration of 200 mM to achieve an arrest in S-phase;

nocodazole (Sigma M1404) was added to a final concentration of 5 µg/ml to achieve an arrest in M phase. Cell cycle arrest was confirmed by using a microscope and by taking samples for flow cytometry. To release cells from a cell cycle arrest, cells were washed once with and then re-suspended in pre-warmed YP-medium containing the appropriate sugar. To deplete cells of a protein carrying an auxin-inducible degron (AID) tag, indole-3-acetic acid (IAA, Sigma I3750) was added to 3 mM final concentration. A pre-treatment of cells with doxycycline was required to allow effective depletion via the iAID-system[79]. Specifically, *sld3-iAID* cells were cultured in the presence of 0.1 µg/ml doxycycline (DOX, Sigma D9891) and the doxycycline concentration was increased to 20 µg/ml when IAA was added. EdU (Santa Cruz Biotechnology sc-284628) was used at a final concentration of 100 µM to label newly synthesized DNA.

### Plasmids
Genes of interest were amplified from genomic DNA of W303-1A and cloned into the respective vector using the In-Fusion HD cloning kit (Clontech). Mutations and deletions were introduced by oligonucleotide-directed site-specific mutagenesis. All plasmids used in this study are listed in Supplementary Table 2.

### Flow cytometry
About $10^7$ cells (0.5–1 OD) were harvested by centrifugation, resuspended in 50 mM Tris-HCl pH 8.0/70% ethanol, and stored at 4 °C for at least 1 h for fixation and permeabilization. Afterwards cells were digested with RNaseA buffer (50 mM Tris-HCl pH 8.0, 0.38 mM $MgCl_2$, 0.38 mg/ml RNase A (Sigma R4875)) overnight at 37 °C and with proteinase K buffer (50 mM Tris-HCl pH 8.0, 5% glycerol, 2.5 mM $CaCl_2$, 1 mg/ml proteinase K (Sigma P2308)) for 30 min at 50 °C. Cells were resuspended in 50 mM Tris-HCl pH 8.0, sonicated, diluted 1:20 with 50 mM Tris-HCl pH 8.0 containing 0.5 µM SYTOX green (Invitrogen S7020), and measured on a MACSquant analyzer (Miltenyi Biotec).

To measure DNA synthesis via flow cytometry, EdU-treated cells were processed analogously to samples for cell cycle analysis and afterwards incubated for 60 min in PBS supplemented with 1% BSA. One half was subjected to a click chemistry reaction with disulfo-Cy5-picolyl-azide (Jena Bioscience CLK-1177) for 1 h, whereas the other half was kept as a control. A click chemistry reaction for $10^7$ cells (1 OD) consisted of 36 µl PBS, 2 µl freshly prepared 1 M ascorbic acid, 2 µl 1 M $CuSO_4$, and 0.5 µl 2 mM disulfo-Cy5-picolyl-azide. After the click chemistry reaction, the cells were washed twice with 10% ethanol in PBS and resuspended in PBS. Both the click chemistry reaction and the control samples were diluted 1:20 with SYTOX buffer (50 mM Tris-HCl pH 8.0, 0.5 µM SYTOX green) and measured on a MACSquant analyzer (Miltenyi Biotec).

Flow cytometry data were analyzed and plotted using FlowJo (v10.6.2). For quantification, B1 channel (SYTOX green fluorescence) measurements were exported and fitted to a bimodal distribution model with one population anchored on the 1 C DNA content peak using the package mixtools (v1.2.0)[107] in R (v4.0.3).

### EdU-IP for sequencing
For each sample, ~$10^9$ cells (100 OD) were harvested by centrifugation, fixed with 50 mM Tris-HCl pH 8.0/70% ethanol for at least 1 h, and then digested with 25 ml RNaseA buffer (50 mM Tris-HCl pH 8.0, 0.38 mM $MgCl_2$, 0.38 mg/ml RNase A) overnight at 37 °C. Afterwards, cells were washed with 50 mM Tris-HCl pH 8.0, digested with 10 ml proteinase K buffer (50 mM Tris-HCl pH 8.0, 5% glycerol, 2.5 mM $CaCl_2$, 1 mg/ml proteinase K) for 1 h at 50 °C, and subsequently incubated with 25 ml PBS supplemented with 1% BSA for another hour at room temperature. The cells were afterwards subjected to an upscaled click chemistry reaction with biotin-picolyl-azide (Jena Bioscience CLK-1167) for 1 h at room temperature and washed twice with 10% ethanol in PBS afterwards. Next, cells were resuspended in breaking buffer (2% triton X-

100, 1% SDS, 100 mM NaCl, 10 mM Tris-HCl pH 8.0, 1 mM EDTA) and subjected to mechanical lysis. DNA from this lysate was sheared to 300 bp fragments using a BioRuptor UCD-200 sonicator (Diagenode). Cell debris was removed by high-speed centrifugation and DNA from the supernatant was isolated by ethanol precipitation and resuspension in TE buffer. Labeling of the DNA with biotin was confirmed in dot blots using HRP-conjugated streptavidin (Sigma S5512, 1 µg/ml) for detection. The size distribution of DNA fragments was analyzed by agarose gel electrophoresis and 20 µl were taken aside for library preparation (total DNA).

Equal amounts (approx. 700 ng) of sheared, EdU-biotin-labeled DNA were mixed 1:1 with 2× WB buffer (10 mM Tris-HCl pH 8.0, 10 mM EDTA, 1 M NaCl, 0.02% NP-40) supplemented with 1 mg/ml BSA and incubated with 25 µl of streptavidin-coupled magnetic beads (Thermo Fisher Scientific Dynabeads M-280) for 30 min at room temperature. The beads were washed five times for 5 min with 1× WB buffer (diluted with TE buffer). Subsequently, the beads were eluted twice with 100 µl buffer EB (10 mM Tris-HCl pH 8.0, 10 mM EDTA, 0.1% SDS) for 1 h at 55 °C. The eluates were pooled, purified by phenol/chloroform/isoamyl alcohol extraction, and precipitated in the presence of 50 µg/ml Glyco-Blue coprecipitant (Invitrogen AM9515) with sodium acetate and absolute ethanol. After drying, the pellet was resuspended in 20 µl TE buffer.

Libraries for next-generation sequencing were prepared using the NEBNext Ultra II DNA library prep kit (New England Biolabs) following the manufacturer's instructions and sequenced on an Illumina NextSeq 500 platform (75 bp reads, paired-end) at the MPIB NGS core facility.

### Western blots
Approximately $2 × 10^7$ cells (1 OD) were harvested by centrifugation and snap-frozen in liquid nitrogen. Afterwards, cells were resuspended in 1 ml water, supplemented with 150 µl 1.85 M NaOH and 7.5% β-mercaptoethanol, and incubated for 15 min at 4 °C. Subsequently, 150 µl 55% tri-chloroacetic acid (TCA) were added for 10 min at 4 °C, before collecting the pellet, resuspending it in 50 µl HU buffer (8 M urea, 5% SDS, 200 mM Tris-HCl pH 6.8, 1.5% DTT, bromophenolblue), and heating it for 10 min at 65 °C.

Samples were loaded on NuPAGE 4–12% Bis-Tris acrylamide gels (Invitrogen NP0322) and run at 200 V with MOPS buffer or MES buffer, according to the proteins that needed to be separated. To resolve phosphorylated isoforms of Rad53, standard 10% acrylamide gels were run with SDS buffer.

After gel electrophoresis, proteins were transferred to nitrocellulose membranes using a tank blot system and methanol-containing transfer buffer. The transfer was carried out at 4 °C with 90 V for 90 min. After transfer, primary antibodies were diluted in superblotto (2.5% skim milk powder, 0.5% BSA, 0.5% NP-40, 0.1% tween-20 in TBS) and added to the membranes for incubation overnight at 4 °C. After washing once for 5 min with western wash buffer, secondary antibodies (diluted 1:3000 in superblotto) were added for 90 min at room temperature. For detection of the immune-blots, Pierce ECL western blotting substrate (Thermo Fisher Scientific 32106) was added following the manufacturer's instructions and chemiluminescence was detected using a LAS-3000 CCD camera system (Fujifilm). Images of uncropped western blots are presented in Supplementary Fig. 8.

Primary antibodies used in this study are: anti-γH2A (abcam ab181447, rabbit, 1:2000 dilution), anti-Rad53 (abcam ab104232, rabbit, 1:4000 dilution), anti-Dpb11 (BPF19[71], rabbit, 1:5000 dilution), anti-Sld2 (kind gift of Philip Zegerman, rabbit, 1:2000 dilution). A polyclonal HRP-coupled anti-rabbit IgG (Jackson Immuno Research 111-035-045, goat, 1:3000) was used as the secondary antibody.

### Immunoprecipitation of replisomes
Experiments were done as triplicates and, for each sample, $2 × 10^9$ cells (100 OD per sample) were stopped by the addition of 0.1% $NaN_3$ and

kept on ice for 30 min before being harvested by centrifugation. Replisomes were purified based on a previously published work[52]. Briefly, cells were washed with 10 mM HEPES-KOH pH 7.9, resuspended in lysis buffer (100 mM HEPES-KOH pH 7.9, 50 mM potassium acetate, 10 mM magnesium acetate, 2 mM EDTA) including protease inhibitors, and snap-frozen as yeast popcorn in liquid nitrogen. The yeast popcorn was ground to a fine powder using a cryogenic mill (SPEX SamplePrep), thawed, and supplemented with 0.25 volumes of glycerol mix buffer (100 mM HEPES-KOH pH 7.9, 300 mM potassium acetate, 10 mM magnesium acetate, 2 mM EDTA, 50% glycerol, 0.5% NP-40) to obtain an extract with 10% glycerol, 100 mM potassium acetate and 0.1% NP-40. After incubation with 800 U/ml SmDNase (MPIB core facility) for 30 min on ice, the extract was cleared by centrifugation. The protein concentration was measured using a standard Bradford assay and, after adjusting the concentrations, the extracts were used directly for immunoprecipitation.

Agarose GFP-trap beads (Chromotek gta-100, 20 µl used per sample) were equilibrated with IP wash buffer (100 mM HEPES-KOH pH 7.9, 100 mM potassium acetate, 10 mM magnesium acetate, 2 mM EDTA, 10% glycerol) including 0.1% NP-40 and then incubated with 30 mg of total protein for 2 h at 4 °C. Afterwards, the beads were washed three times with IP wash buffer including NP-40 and two times with IP wash buffer lacking NP-40.

## Mass spectrometry measurement

Washed beads were incubated for 30 min with elution buffer 1 (2 M urea, 50 mM Tris-HCl pH 7.5, 2 mM DTT, 20 µg/ml trypsin) followed by a second elution with elution buffer 2 (2 M urea, 50 mM Tris-HCl pH 7.5, 10 mM chloroacetamide) for 5 min. Both eluates were combined and further incubated at room temperature overnight. Tryptic peptide mixtures were acidified to 1% TFA and desalted with Stage Tips containing C18 reverse-phase material and analyzed by mass spectrometry.

Peptides were separated on 50 cm columns packed with ReproSil-Pur C18-AQ 1.9 µm resin (Dr. Maisch GmbH). Liquid chromatography was performed on an EASY-nLC 1200 ultra-high-pressure system coupled through a nano-electrospray source to a Q-Exactive HF-X Mass Spectrometer (Thermo Fisher Scientific). Peptides were loaded in buffer A (0.1% formic acid) and separated with a non-linear gradient of 5–60% buffer B (0.1% formic acid, 80% acetonitrile) at a flow rate of 300 nl/min over 50 min. The column temperature was kept at 60 °C by an in-house designed oven with a Peltier element. Data acquisition switched between a full scan (60 K resolution, 20 ms max. injection time, AGC target 3e6) and 10 data-dependent MS/MS scans (15 K resolution, 60 ms max. injection time, AGC target 1e5). The isolation window was set to 1.4 and normalized collision energy to 27. Multiple sequencing of peptides was minimized by excluding the selected peptide candidates for 30 s.

## Pulsed-field gel electrophoresis and Southern blotting

Pulsed-field gel electrophoresis and Southern blotting were performed with modifications as previously described[108]. For each timepoint, ~4 × 10⁷ cells (2 OD) were harvested, resuspended in ice-cold Stop Buffer (150 mM NaCl, 50 mM NaF, 2 mM NaN₃, 10 mM EDTA), and stored at 4 °C until further processing. Samples were washed twice with ice-cold 50 mM EDTA, resuspended in SCE buffer (1 M sorbitol, 0.1 M sodium citrate, 10 mM EDTA) + 150 U/ml zymolyase 100 T (Roth 9329), and mixed with 50 µl 2% agarose before casting into plugs. After solidification, plugs were placed in SCEM (1 M sorbitol, 0.1 M sodium citrate, 10 mM EDTA, 5% β-mercaptoethanol) + 150 U/ml zymolyase 100 T and incubated at 37 °C for 2 days. Afterwards, plugs were washed with TE (10 mM Tris-HCl pH 8.0, 1 mM EDTA) for 1–2 h each wash, placed into PK buffer (1 mg/ml sarcosyl, 0.5 M EDTA, 2 mg/ml proteinase K), and incubated at 55 °C for 2 days. Plugs were washed three more times with TE before use.

Plugs were loaded on a gel containing 1% agarose (Bio-Rad Cat. 1620138) in 0.5× TBE (45 mM Tris, 45 mM borate, 0.5 mM EDTA). Electrophoresis was carried out in 14 °C cold 0.5× TBE in a CHEF DR-III system (Bio-Rad, initial switch time 60 s, final switch time 120 s, 6 V/cm, angle 120°, 24 h). Afterwards, the gel was stained with 1 µg/ml ethidium-bromide in 0.5× TBE for 1 h and de-stained with deionized water. Images were taken using a GenoSmart gel documentation system (VWR).

For Southern blotting, the DNA was nicked in 0.125 M HCl for 10 min, denatured in 1.5 M NaCl, 0.5 M NaOH for 30 min, and neutralized by 0.5 M Tris, 1.5 M NaCl (pH 7.5) for 30 min. The DNA was transferred onto a Hybond-N + membrane (GE healthcare) and UV-cross-linked (Stratagen Stratalinker 1800, auto-crosslink function). The membrane was probed with a radioactive (α-³²P dCTP) labeled TRP1 fragment and imaged using a Typhoon FLA 9000 imaging system (GE Healthcare).

## Strand-specific RPA-ChIP-seq

Samples for strand-specific RPA-ChIP-seq were prepared as described previously[81]. Briefly, 2 × 10⁹ cells (100 OD) were crosslinked at the indicated timepoints with 1% formaldehyde for 16 min at room temperature, subsequently quenched with 400 mM glycine for 60 min, washed with PBS, and frozen in liquid nitrogen. After resuspending in lysis buffer (50 mM HEPES-KOH pH 7.5, 150 mM NaCl, 1 mM EDTA, 1% triton X-100, 0.1% sodium-deoxycholate, 0.1% SDS), cells were mechanically lysed and chromatin was sheared to 200-500 bp fragments. Cell lysates were cleared by centrifugation and diluted 1:1 with lysis buffer. 1% of the extract was taken as a total DNA sample, and 40% of the extract were incubated with an antibody against budding yeast RFA (Agrisera, AS07 214) for 2 h followed by 30 min incubation with protein A-coupled dynabeads (Invitrogen 10002D). Beads were washed three times with lysis buffer, once with lysis buffer supplemented with 500 mM NaCl, once with wash buffer (10 mM Tris-HCl pH 8.0, 0.25 M LiCl, 1 mM EDTA, 0.5% NP-40, 0.5% sodium-deoxycholate), and once with TE pH 8.0. Immunoprecipitated complexes were eluted with 1% SDS, proteins were degraded with proteinase K, and crosslinks were reversed at 65 °C. DNA was purified by phenol-chloroform extraction and cleaned up using Phase Lock Gel tubes (5Prime) and ethanol precipitation.

Strand-specific ChIP-seq libraries were prepared from 1 to 3 ng of DNA using Accel-NGS 1 S Plus Library Kit (Swift Biosciences) following the manufacturer's instructions and sequenced on an Illumina NextSeq 500 (75 bp or 37 bp reads, paired-end) at the MPIB NGS core facility.

## Split-Venus fluorescence intensity measurement and quantification

Cells were grown at 30 °C in YPD supplemented with adenine to log-phase (OD600 of 0.5–0.6), stopped by adding 0.1% NaN₃, and kept in the dark on ice for 30 min. After two washes with 50 mM Tris-HCl pH 8.0, cells were re-suspended in 50 mM Tris-HCl pH 8.0 and measured on a MACSquant analyzer (Miltenyi Biotech). Values for mean as well as 97th percentile fluorescence intensity were calculated and exported using FlowJo (v10.6.2) and data from 6 independent cultures per strain were used to generate boxplots using R (v4.0.3) after subtracting background fluorescence as measured in a strain that expressed an untagged SLD2 construct.

## Gross chromosomal rearrangement assay and rate calculation

Rates of gross chromosomal rearrangements (GCRs) were determined using a standard protocol[85]. Briefly, pre-cultures of S. cerevisiae cells harboring a CAN1::URA3 reporter on chromosome 5 were grown in SC-Ura medium and plated out on YPD plates to obtain colonies that formed from single cells. Eight colonies were excised from the plates for each condition and used to inoculate larger cultures in YPD (control strains: 50 ml; strains with stabilized interaction: 2 ml), which were

grown to stationary phase at 30 °C. The number of viable cells was determined by plating a serial dilution ($10^{-6}$) on non-selective YPD plates. The total number of GCR events was determined by plating the remaining culture on SC-Arg plates that were supplemented with 50 mg/L L-canavanine (Sigma C9758) and 1 g/L 5′-fluoroorotic acid (US Biological Life Sciences F5050) to select against both *CAN1* and *URA3*. No more than $10^9$ cells were spread on each selection plate and the plates were incubated at 30 °C for 2 days (YPD) and 3–5 days (selection). Afterwards, the clones were counted and GCR rates as well as 95% confidence intervals were calculated by fluctuation analysis using the maximum likelihood method in the web tool FALCOR[109] that was kindly made accessible by the Liang lab under https://lianglab.brocku.ca/FALCOR/ (no version data available, as used in September 2020).

### Mass spectrometry data analysis

Raw mass spectrometry data were analyzed with MaxQuant (v1.5.3.54)[110]. Peak lists were searched against the yeast Uniprot FASTA database combined with 262 common contaminants by the integrated Andromeda search engine. The false discovery rate was set to 1% for both peptides (minimum length of 7 amino acids) and proteins. "Match between runs" (MBR) with a maximum matching time window of 0.5 min and an alignment time window of 20 min was enabled. Relative protein amounts were calculated with the MaxLFQ algorithm with a minimum ratio count of two.

Absolute protein intensity (iBaq) estimates were calculated dividing the LFQ intensities by the theoretical number of tryptic peptides of each protein[111].

Statistical analysis of LFQ-derived protein expression data was performed using R. LFQ values were log2 transformed. Per each experimental condition, a pairwise comparison was performed with triplicate pulldowns of bait (GFP-Psf2) versus control (untagged Psf2) yeast strains. For each comparison, the dataset was filtered to present at least two valid values in the bait group and missing values imputed with a downshift of 1.8 standard deviations and a width of 0.2 standard deviations. Fold changes and $p$ values were calculated using an unpaired Student's *T* Test.

### Estimation of the average speed of a replisome in G1

To estimate the average speed of a replisome in G1, we quantified flow cytometry data for control and CDK/DDK-bypass strains (Fig. 3d). The increase in mean SYTOX green fluorescence for the CDK/DDK bypass strain per hour was corrected for background DNA synthesis (mtDNA replication) using the control strain and normalized to the mean SYTOX green fluorescence at t = 0 h to yield a bulk replication speed. The bulk replication speed of the CDK/DDK-bypass strain in G1 was 0.25 C/h. This means that a full round of replication (1 C of DNA) using this system takes 4 h = 240 min. For the calculation, we considered the time frame of 210–270 min since we harvested samples only once per hour. The haploid budding yeast genome contains ~12,000 Kb of DNA, so this yields 12,000 Kb/240 min = 50 Kb/min (44–57 Kb/min for the considered time frame for G1 replication) as a bulk replication speed in G1. Replication in S phase takes about 25 min, so we get a bulk replication speed of 12,000 Kb/25 min = 480 Kb/min.

There are ~400 origins of replication in budding yeast and we assume that 40% = 160 origins are active in early S-phase, each giving rise to 2 replisomes traveling in opposite directions, so 320 replisomes in total. Our quantitative mass spec data (Fig. 2d) indicated that there are about 8× more replisomes present during replication in early S phase compared to G1 replication using the CDK/DDK-bypass strain. Fold-changes for individual sub-units gave a range of 5.5x–10.3x for the difference in abundance of replisomes in G1 and S. Based on these data, we estimated that there are 40 replisomes (31-57 replisomes when considering extreme values measured for the individual sub-units) present per cell during G1 replication.

We estimated the average speed of a replisome by dividing the speed of bulk replication by the number of replisomes. For a replisome in S-phase, we estimate an average speed of 1.5 Kb/min (=480 Kb/min/320 replisomes); For a replisome in G1, we estimate an average speed of 1.25 Kb/min (=50 Kb/min/40 replisomes) that falls within 0.8–1.8 Kb/min (=44 Kb/min/57 replisomes as a lower boundary and 57 Kb/min/31 replisomes as an upper boundary).

### Transcriptome analysis

For each sample, ~$10^8$ cells (10 OD) were harvested by centrifugation and flash-frozen in liquid nitrogen. Total cellular RNA was isolated from these samples using MasterPure Yeast RNA Purification Kit (MPY03100) according to the manufacturer's instructions, including digestion of genomic DNA with DNaseI.

Libraries for next-generation sequencing were prepared from 1 μg of total RNA using the NEBNext Ultra II Directional RNA library prep kit for Illumina (E7765, New England Biolabs) with NEBNext Poly(A) mRNA Magnetic Isolation Module (E7490, New England Biolabs) following the manufacturer's instructions. A fresh 1:100 dilution of ERCC RNA Spike-In Mix (4456740, Invitrogen) was added to the total RNA before library preparation. Sequencing was performed on an Illumina NextSeq 500 platform (42 bp reads, paired-end) at the MPIB NGS core facility.

Sequencing reads (about 10 million per sample) were mapped to the S. cerevisiae reference genome (sacCer3) using the STAR aligner (v 2.7.10a)[112] with default parameters and converted to sorted BAM files using samtools (v 1.12). We used htseq-count (v 2.0.1)[113] to count reads per transcript with the parameters "--mode=intersection-nonempty --order=pos --stranded=reverse --nonunique=none --idattr=gene_name --add-chromosome-info". MA plots and PCA analysis were performed using DESeq2 (v 1.36.0)[114] with standard parameters based on the raw count matrices from htseq-count following the instructions in the package's vignette. To analyze expression levels of individual transcripts, we first calculated RPK (reads per kilobase) values by dividing the number of counts by the length of the gene (in Kb). The TPM (transcripts per million) values were then calculated by dividing the RPK value by a scaling factor (sum of all RPK values in the sample, divided by one million).

### Next-generation sequencing data analysis

For each sample, about 10 million sequencing reads were obtained and quality-checked using FastQC (v0.11.9, https://www.bioinformatics.babraham.ac.uk/projects/fastqc/). The reads were aligned to the budding yeast reference genome sacCer3[115] using the Burrows-Wheeler aligner bwa (v0.7.17) with standard parameters[116] and the alignments were sorted and indexed using samtools (v1.12)[117]. The tool bamCoverage from deepTools (v3.5.1)[118] was used with the options "--binSize 50 --minMappingQuality 60 --normalizeUsing CPM" to calculate the coverage of uniquely mapping reads per 50 bp bin and normalize it to sequencing depth as counts per million mapped reads (CPM). Reads mapping to the rDNA locus were blacklisted during this step. In a second step, we used the tool bigwigCompare from the deepTools suite to normalize samples to total input DNA by calculating the ratio of EdU-labelled DNA or RPA-bound DNA to total DNA using larger bin sizes (1 kb–2 kb, depending on the experiment). Locations of origins of replication were used as annotated in oriDB[119] and annotations for telomeres and centromeres were taken from SGD[120]. Data were plotted using plotHeatmap from the deepTools suite or pyGenomeTracks (v3.6)[121].

To separate reads by strands, alignment files were filtered with samtools using the options "-f 99" and "-f 147" for reads mapping to the forward strand and the options "-f 83" and "-f 163" for reads mapping to the reverse strand before calculating bigWig-coverage files. To calculate asymmetry profiles, the log2 ratio of depth-normalized forward and reverse reads was first calculated and then averaged over all

chromosomes using computeMatrix and plotHeatmap/plotProfile from deepTools.

To calculate a normalized single-stranded DNA asymmetry score per chromosome, the average absolute value of asymmetry (log2(forward/reverse)) per 50 bp bin was calculated and plotted against total chromosome length using R (v4.0.3).

## Comparing origin usage in S phase with or without prior replication in G1

Peaks corresponding to origins of replication were called using MACS2 (v 2.2.7.1)[122] function "callpeak" by comparing alignments of EdU-IP data to the corresponding total DNA control with default parameters in paired-end mode with a $q$ value of 0.1. The positions of the summits of the peaks were extended by 5 Kb in both directions and pairwise overlaps between these peak regions were determined using bedtools intersect (v 2.30.0)[123] and depicted as a Venn diagram.

## Analysis of survivors of G1 replication

Single clones were grown at 30 °C in 5 ml YPD to stationary phase. Cell pellets were resuspended in breaking buffer (2% triton X-100, 1% SDS, 100 mM NaCl, 10 mM Tris-HCl pH 8.0, 1 mM EDTA) and mechanically lysed. DNA from this lysate was purified by phenol-chloroform-extraction and ethanol precipitation. Sequencing libraries were prepared using the DNA PCR-Free Prep Tagmentation kit (Illumina, 20041795) following the manufacturer's instructions and sequenced on an Illumina NextSeq 500 platform (75 bp reads, paired-end) at the MPIB NGS facility.

Sequencing reads were aligned to the budding yeast genomes (sacCer3) with bwa mem using default parameters. Copy number variations were calculated directly from the alignment-files using FREEC (v 11.6)[124]. Data were binned at 1000 bp and mitochondrial DNA was excluded from the analysis. Standard ploidy was set to 1 and the parameter for pre-telomeric/pre-centromeric regions was set to 20,000 bp.

## Reporting summary

Further information on research design is available in the Nature Research Reporting Summary linked to this article.

## Data availability

The sequencing data generated in this study have been deposited in NCBI's Gene Expression Omnibus[125] under accession number GEO Series accession number GSE208590. The mass spectrometry proteomics data have been deposited to the ProteomeXchange Consortium via the PRIDE[126] partner repository with the dataset identifiers PXD028308 and PXD035629. Source data are provided with this paper.

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

## Acknowledgements

We thank Uschi Schkölziger and Sandra Mitzkus for excellent technical assistance, Helle Ulrich, Etienne Schwob, Seiji Tanaka for yeast strains and plasmids, Jonathan Baxter, Armelle Lengronne, Philippe Pasero, Iestyn Whitehouse for protocols and/or experimental advice, Marja Driessen, Rin ho Kim (MPIB NGS core facility) for next-generation sequencing, Stefan Pettera (MPIB core facility) for peptide synthesis, Guy Riddihough, Life Science Editors for editing the paper, Dominik Boos, Stephan Hamperl, Christoph Kurat and all members of the Pfander lab for stimulating discussion and critical reading of the paper. This work was supported by the Max Planck Society (to B.Pf., to M.M.), grants by the Deutsche Forschungsgemeinschaft (DFG, German Research Foundation): PFA794-5/1 (to B.Pf.); Project-ID 213249687 – SFB 1064 (A23 to B.Pf.) and by the Fondation ARC pour la Recherche sur le Cancer (ARCPJA22020060002119 to BPa).

## Author contributions

K.U.R. and B.Pf. conceived and designed the study. K.U.R. and J.B. conducted experiments, M.P. prepared libraries for strand-specific RPA-ChIP-seq, M.W. measured and analyzed mass spectrometry data, K.U.R. analyzed NGS data. M.C. and B.Pa. helped with DNA fibre experiments during the revision of the paper. K.U.R. and B.Pf. wrote the paper. All authors analyzed data and commented on the paper. M.M. and B.Pf. secured funding.

## Funding

## Competing interests

The authors declare no competing interests.
