## [Peer Review File · Nature Communications]

Unscheduled DNA replication in G1 causes genome instability and damage signatures indicative of replication collisionsREVIEWER COMMENTS

Reviewer #1 (Remarks to the Author):

This submission reports the outcomes and modifiers of an experimental system in which DNA replication is induced by circumventing the controls that typically delay the onset of S-phase in yeast. Specifically, as shown previously, the onset of DNA synthesis requires both cyclin dependent kinase (CDK) and Dbf4-Dependent Kinase (DDK), and in the reported system replication is induced in yeast harboring an inactive (kept in check by the consistent presence of alpha factor) by inducing the expression of active forms of the CDK downstream targets the catalytic subunit of DDK. This system facilitates the identification of potential modifiers that affect the cellular networks that normally prevent DNA replication during the G1 phase of the cell cycle. The observations are potentially important because the correct timing of chromosome duplication is essential for normal homeostasis, including during development, and because unscheduled DNA replication is also considered as a target in cancer therapy as it is known to occur at the early stages of tumorigenesis and can drive cell death in cells that do not trigger the appropriate cell cycle checkpoints.

The reported study demonstrates that DNA synthesis induced by bypassing S-phase kinase regulation (early-onset replication) starts at the same replication origins ("canonical origins") that initiate DNA replication during S-phase although the replisomes catalyzing early-onset replication differ in some components from replisomes utilized during normal S-phase. The paper also reports a series of experiments aimed to identify cellular factors that limit early-onset replication, providing evidence for a role of histone availability. If the cells are subsequently allowed to activate CDK (release from alpha factor block), further replication could be deleterious, forming long tracks of single stranded DNA and leading to DNA breakage and chromosomal aberrations.

This is a clearly written paper addressing an important problem with appropriate experimental tools. The experimental system can potentially provide insights into the essential processes that regulate chromosome duplication. In its current form, the paper will benefit from a few clarifications of the system and a detailed discussion of some aspects of the findings, as detailed below.

Comments and suggestions:

- The current submission defines the induced early-onset replication as "replication in G1" because cells start replication while CDK activity is blocked by alpha factor. This term can be confusing, however, because although CDK activity is indeed lower in the G1 phase, cell cycle phases are broadly defined with respect to DNA synthesis and not kinase activity (for example, G1 is defined as the growth period prior to the onset of DNA synthesis). Other studies in which DNA replication is induced early usually refer to "early S-phase entry", can this definition be used in this case? Alternatively can the authors simply refer to "DNA replication in the absence of CDK activity"?
- The FACS images in figures 1 and 3 suggest that cells with early-onset replication undergo some over-replication. It would be useful to provide quantitative data about the extent of re-replication, which genomic regions are over-represented, and discuss similarities and differences between DNA synthesis induced by kinase bypass and other instances of early-onset replication.
- Related to the above, the paper provides strong evidence that the same replication origins are involved in initiation of DNA synthesis regardless of whether replication starts normally during S-phase or is induced in the absence of CDK activity. Less information is provided about the utilization of replication origins during replication subsequent to alpha factor removal (activation of CDK) - are the same origins re-replicated and to which extent. This is important because CDK activity is known to regulate licensing (specifically prevent the re-association of pre-replication complexes with chromatin during S-phase) so it would be interesting to ask directly if the early-onset replicating origins are still licensed.
- The paper provides data suggesting that replisomes activated in the absence of CDK activity differ from replisomes activated during normal S-phase. It would be interesting to learn how replisome

composition (specifically, the abundance of CMG) in cells undergoing early-onset replication would be affected by the release from alpha factor. For example, if low ratios of MCM10 persist upon release from alpha factor, could faster replication with low-MCM10 replisomes underly the observed DNA damage?

- If the model in Figure 7 is correct, allowing early-onset replication forks to terminate before the release from alpha factor (for example by terminating the overexpression of Dbf4 and Dbf11) should prevent DNA breakage. Can this be tested?

Minor

- Figure 1B, left panel, bottom row: EdU incorporation is not evident in the asynchronous cell population, although this population should contain S-phase cells. Why?

Reviewer #2 (Remarks to the Author):

GENERAL COMMENTS

In this work, Reusswig and colleagues used budding yeast as a model to study the unscheduled replication in the G1-phase and its consequence in the following S-phase cells. The authors engineered genetic systems in budding yeast to generate strains inducing unscheduled replication in the G1-phase, and quantified the composition of replisomes in G1- and S-phase by quantitative proteomics analysis and showed some interesting differences between G1 replisome and S-phase replisome. The authors further showed that G1 replication per se did not trigger cellular checkpoints, while subsequent replication during S-phase of cells with unscheduled replication in G1 lead to over-replication, chromosome breaks and cell cycle arrest. Strikingly, the authors observed a strong strand-biased occurrence of RPA-bound single-stranded DNA in the subsequent S-phase cells and suggest a model in which head-to-tail replication fork collisions between S-phase replication forks and G1 replication forks to explain their observation.

The authors used both molecule biology and genome-wide approach to study the unscheduled replication in budding yeast cells. The authors performed considerable works with engineering numerous yeast strains bypass CDK and DDK control to characterize the unscheduled G1 replication and its consequences. The manuscript is well structured and written. In my opinion, it's a well-performed study, which provides interesting results for researchers working in the field of DNA replication stress and genome instability. However, some control experiments need to be added to draw a solid conclusion.

SPECIFIC COMMENTS:

Major comments:

- In several places, the authors indicated that the unscheduled G1 replication progresses slower than canonical S-phase replication, e.g. Page 3, the last paragraph of Introduction, Page 13, Discussion "When compared to replication in S-phase, replication in G1 progressed approximately tenfold slower in bulk.... We show that inefficient replication initiation and inefficient replication elongation both contribute to overall slow replication (Fig. 3)." However, when I checked Fig. 3, it's not clear whether it's only due to the amount of initiation (or ongoing replication forks) lower in G1 than S or it's also due to the replication fork speed is slower in G1 replication. The authors should clarify this point. And also, they might need to perform some additional experiments, such as fibre assay to check replication fork speed to make a clearer and stronger conclusion.

Another point is that: do the transcriptional stage, which might can be different between the G1 and S

cells (e.g. higher transcription in G1 than S), might also contribute to the observed difference between G1 and S phase replication?

- Relative to the previous point, in the model suggested in Fig. 7, the authors assume that the S-phase forks go faster to catch up with the G1 forks, which leads to head-to-tail fork collision. However, it's not clear why the forks resulted from the G1 initiation cannot reach the same speed as the forks from the S phase initiation once cells enter S-phase.

- Although I'd like to trust the results of Western blots shown by the authors, without any internal loading control shown together with the figures (Fig 4 and Fig S4), it's really hard to judge the quality of the results. For example, in Fig. 4D, there is a clear variation amongst samples, for instance, Ddc1-Rad9 3h G1, 20, 40 release samples, show a much lower level of Rad53 compared to the same time points of Ddc1 samples. The authors need to provide Western plots results with proper controls.

Also, please change the "control" to "control cells" in all figures (at least the Western plots ones) to avoid any confusion.

- I think the asymmetry of RPA-ChIP-seq pattern observed by the authors in the current study is the most striking result. The authors did a good job to show that such an asymmetry pattern is unlikely due to BIR. However, it's not clear at all, how single-ended DSBs resulted from the head-to-tail fork collisions (if it indeed happens, see the previous comment) can explain the observed pattern.

For example, in Fig. 7 B and C, it's not clear which strand is associated with RPA. It will be great to add the RPA on the figures to make it clear. In addition, the authors should make a clearer scheme to explain how single-ended DSBs cause by the head-to-tail fork collisions can explain the observed asymmetry RPA pattern.

- Page 1, Abstract, "resulted in over-replication and led to chromosome breaks via head-to-tail replication fork collisions that are marked by chromosome-wide, strand-biased occurrence of RPA-bound single-stranded DNA." Since the head-to-tail replication fork collision is just a model and not demonstrated by any experiment shown in the current manuscript, the authors should not provide such a strong statement in the abstract. The authors might also need to modify their title to better describe what they indeed observe in their study. Or not, the authors need to provide stronger evidence of head-to-tail fork collision to support their statement.

Minor comments:

- At the beginning of the Result section, please explain why the CDK and CDK/DDK bypass will not trigger the premature S-phase entry in the experimental design. Although the authors explain in detail their experimental design in the Method, it's good to clear explain it too, at least at the beginning of the Result section, which will be useful for the people not working on DNA replication in yeast.

- In general, the figure legends contain limited information. Please provide more details in the figure legends, which will be very useful for the readers to better understand the figures and do not need to

check forward and backward in the text and other figures in order to fully understand.

- Fig. 1C, on the Y-axis, it indicates "coverage EdU-IP", is it a problem of legend? Or not why is it not the same as the other figures using "coverage EdU-IP/Input"?

Also, please provide the Y-axis scale. The same for all other figures showing the genome-wide profiles.

- Fig. 1D, for the cluster 1 early ARS, the EdU-IP in CDK/DDK bypass (Dbf4_Dpb11) is much lower than the CDK (Dpb11) bypass. Can the authors provide an explanation?

- Fig. 2C, G1 replication, please specify which strain was used in the analysis, is it CDK bypass or CDK/DDK bypass. Is there any difference between them?

- Fig. 1 D and Fig. 5E: although, in Fig 1D, the EdU-seq data show a big difference between Dpb11 (only peak around cluster 1 early ARS and not cluster 2) and Dbf4_Dpb11 (peaks on both cluster 1 and cluster 2, and the difference between the two clusters is much smaller), the read asymmetry on RPA-ChIP-seq shown in Fig 5E is very similar for Dpb11 and Dbf4_Dpb11 (e.g. cluster 1 is much stronger than cluster 2). Can the authors comment on that?

- Also, in Fig. 5E, the profiles of RPA-ChIP-seq of 60, 120 and 180 min samples do not seem symmetry around the centres of early ARS, in particular, for the cluster 1. Do the authors have an explanation?

- Page 9, "Both short and long chromosomes were similarly affected, as shown by RPA read asymmetry scores normalized for chromosome length (Supplementary Fig. 5E)." However, when I checked carefully the Fig. 5H, it seems that for later time points (i.e. 120 min and 180 min), the large chromosomes have higher RPA asymmetry scores than the small ones, and it's true for both CDK and CDK/DDK bypass. Please add some quantitative analyses to double check this, and if such correlation is confirmed, please further discuss how to explain the results and whether the results agree with their model.

- Fig. 6D, there are clear peaks observed in all 4 samples at a genomic region, which seem to be an early ARS based on Fig. 1E. Am I correct? Can the author comment on it?

- Page 14, "Due to the bi-directional nature of DNA replication, we think the only theoretical solution to this problem is re-duplication of the entire chromosome." I think the author should be able to test this hypothesis.

Since with the sporadic system (VC-SLD2) that they developed, the authors showed (Fig. S6E,) that the RPA asymmetry was only detected on long yeast chromosomes containing many origins but not the small chromosomes. We should expect to observe the complete duplication only for the long chromosomes but not the small ones at the late S phase. With the whole genome sequencing data (e.g. the input of the RPA-ChIP-seq) of VC-SLD2 cells already available, the authors should be able to test this hypothesis to check whether it's indeed the case or not.

Reviewer #3 (Remarks to the Author):

The manuscript "Unscheduled DNA replication in G1 causes genome instability through head-to-tail replication fork collisions" presents an investigation into the molecular details of early initiation of replication in yeast. The authors used flow cytometry, next-generation sequencing, immunoprecipitation-mass spectrometry (IP-MS) and other techniques to get insight into the process. Since my expertise in the context of the manuscript is limited to mass spectrometry and corresponding data analysis, I will be concentrating solely on the IP-MS experiment.

The authors clearly explain the design of the IP-MS experiment: there are triplicate GFP-Psf2 pulldown samples and triplicate controls with untagged Psf2 per condition. The controls provide the baseline for enrichment analysis for each of the three conditions, as depicted on Fig. 2A, 2B, 2C.

The authors use LFQ intensities to calculate IBAQ values of the replisome-associated proteins, which can be helpful for putting the estimates of the absolute amounts on the same scale. Proteins of the GINS complex have very high and nearly constant IBAQ intensities across the pulldowns (Fig. 2D), which is reasonable since the GFP-tag was attached to the Psf2 subunit of GINS. The further discussion is focused on the relative profiles of intensities of the sub-complexes between the S, S-HU and G1 conditions.

I was able to locate 31 replisome-associated proteins from the Fig. 2 and Supplementary Fig. 2 in the proteomic result table from PRIDE. The proteins of interest were identified with at least 8 unique peptides each, providing a reliable basis for the precursor ion-based quantification. The proteins of interest have either none or 1-2 non-unique peptides each, which simplifies the interpretation of the quantitative results in regards to homology/shared peptides. On the technical side, the LFQ intensities in the protein table show largely unimodal distributions with similar medians and means throughout the range, indicating successful intensity normalisation in MaxQuant. These observations suggest that the relative label-free quantification has a good technical basis.

The authors did not provide the scaling of the IBAQ values to absolute protein concentrations. However, this is inconsequential, as the conclusion about the reduced association of DNA polymerase α /primase with G1 replisomes is based solely on the different magnitudes of relative changes between S and G1.

I would like to suggest a few changes in regards to the reporting of the IP-MS experiment:

- 1) Volcano plots on Fig. 2A-C are depicted with the p-values from Student's t-test. It will not change any of the conclusions of the manuscript, but I would suggest that plotting the adjusted p-values after correction for multiple testing would be more appropriate for this data set.
- 2) Furthermore, it can be possible to use the triplicate data for the replisome proteins and perform statistical testing of the strength of divergence between polymerase α /primase and other components. The resulting p-values could complement the discussion of the Fig. 2D.
- 3) I could not locate a citation regarding IBAQ in the text. Please consider adding a reference, for example, to Schwanhäusser et al. in Nature, 2011 , 473(7347):337-42. doi: 10.1038/nature10098, or to another suitable source of your choice.

Reviewer 1

This submission reports the outcomes and modifiers of an experimental system in which DNA replication is induced by circumventing the controls that typically delay the onset of S-phase in yeast. Specifically, as shown previously, the onset of DNA synthesis requires both cyclin dependent kinase (CDK) and Dbf4-Dependent Kinase (DDK), and in the reported system replication is induced in yeast harboring an inactive (kept in check by the consistent presence of alpha factor) by inducing the expression of active forms of the CDK downstream targets the catalytic subunit of DDK. This system facilitates the identification of potential modifiers that affect the cellular networks that normally prevent DNA replication during the G1 phase of the cell cycle. The observations are potentially important because the correct timing of chromosome duplication is essential for normal homeostasis, including during development, and because unscheduled DNA replication is also considered as a target in cancer therapy as it is known to occur at the early stages of tumorigenesis and can drive cell death in cells that do not trigger the appropriate cell cycle checkpoints.

The reported study demonstrates that DNA synthesis induced by bypassing S-phase kinase regulation (early-onset replication) starts at the same replication origins ("canonical origins") that initiate DNA replication during S-phase although the replisomes catalyzing early-onset replication differ in some components from replisomes utilized during normal S-phase. The paper also reports a series of experiments aimed to identify cellular factors that limit early-onset replication, providing evidence for a role of histone availability. If the cells are subsequently allowed to activate CDK (release from alpha factor block), further replication could be deleterious, forming long tracks of single stranded DNA and leading to DNA breakage and chromosomal aberrations.

This is a clearly written paper addressing an important problem with appropriate experimental tools. The experimental system can potentially provide insights into the essential processes that regulate chromosome duplication. In its current form, the paper will benefit from a few clarifications of the system and a detailed discussion of some aspects of the findings, as detailed below.

We thank the reviewer for the insightful comments and constructive criticism, which we have addressed in their entirety and using new lines of experiments. We are therefore confident that the reviewer will find our paper much improved and suitable for publication.

Comments and suggestions:

R1-1

- The current submission defines the induced early-onset replication as "replication in G1" because cells start replication while CDK activity is blocked by alpha factor. This term can be confusing, however, because although CDK activity is indeed lower in the G1 phase, cell cycle phases are broadly defined with respect to DNA synthesis and not kinase activity (for example, G1 is defined as the growth period prior to the onset of DNA synthesis). Other studies in which DNA replication is induced early usually refer to "early S-phase entry", can this definition be used in this case? Alternatively can the authors simply refer to "DNA replication in the absence of CDK activity"?

We thank the reviewer for bringing up this important point. While we agree that DNA replication is certainly a hallmark of S-phase, it is not the only event of S-phase (centrosome duplication, budding in yeast, etc.). Conversely, DNA synthesis can occur outside of S-phase, as is highlighted for example by the phenomenon of mitotic DNA synthesis (MiDAS), which very clearly occurs in mitosis.

In order to address this point of the reviewer, we tested whether cells would remain in a G1-like state or whether they would be driven in to an S-like state upon induction of replication in this system. To this end, we measured transcriptomes of G1-replicating cells as well as control G1-arrested and S-phase cells. Notably, principal component analysis (PCA) clearly showed “G1-arrested” and “G1 replication samples” to be highly similar and different from S-phase samples (Supplementary Fig. 1c, d). As such we are highly confident that under this condition cells are G1-like and not in S-phase. This condition of “G1 replication” is also clearly different from an “early S-phase entry”, which is for example achieved by Cyclin E overexpression in human cells and generates a premature S-phase state, also from the perspective of cell cycle regulators.

R1-2

- The FACS images in figures 1 and 3 suggest that cells with early-onset replication undergo some over-replication. It would be useful to provide quantitative data about the extent of re-replication, which genomic regions are over-represented, and discuss similarities and differences between DNA synthesis induced by kinase bypass and other instances of early-onset replication.

Following the reviewer’s advice, we added a paragraph to the discussion section of the manuscript outlining similarities and differences of different “early-onset replication systems” such as activation of oncogenic MYC and cyclin E CCNE1 in human cell culture models. We would like to stress however that to our knowledge there is currently no other system to induce “early-onset replication” in yeast.

Furthermore, the reviewer correctly points out that over-replication occurs under conditions of G1 replication. Indeed, we observe cells with $> 2C$ DNA content due to over-replication at late time points of G1 replication (Fig. 1b). We would also like to point the reviewer to our analysis of G1 replication in cells where the licensing factor Cdc6 was depleted. (Supplementary Fig. 4g-h). In this experiment, Cdc6 depletion prevented over-replication and G1 replication for 5 h led to an increase in mean DNA content to only 1.53 C in the absence of Cdc6 instead of 2.19 C in its presence (after subtraction of mitochondrial DNA synthesis seen in control cells, see also Reviewer Figure 1). Importantly, we observed reduced induction of the DNA damage mark γ H2A in Cdc6-depleted cells, altogether showing the extent of over-replication during G1 replication, as well as its contribution to DNA damage induction. To differentiate products of replication and over-replication and to map them to the genome will require development of new technology, which is beyond the scope of the current manuscript.

R1-3

- Related to the above, the paper provides strong evidence that the same replication origins are involved in initiation of DNA synthesis regardless of whether replication starts normally during S-phase or is induced in the absence of CDK activity. Less information is provided about the utilization of replication origins during replication subsequent to alpha factor removal (activation of CDK) - are the same origins re-replicated and to which extent. This is important because CDK activity is known to regulate licensing (specifically prevent the re-association of pre-replication complexes with chromatin during S-phase) so it would be interesting to ask directly if the early-onset replicating origins are still licensed.

We thank the reviewer for bringing up the important point of whether the origin firing program in S-phase is changed by prior G1 replication. To address this point, we conducted a series of new experiments, where we mapped early-firing origins in HU-treated S-phase cells (as peak centres of EdU-labeled replication products) with or without preceding G1. Notably, we find a highly similar pattern of early origin firing under both conditions (Supplementary Fig. 5a, b). From these data we conclude that early firing

origins are efficiently relicensed after G1 replication and that also firing factors appear to operate rather normally in the subsequent S-phase.

As a side observation, we noted a background of DNA synthesis across the entire genome in S-phase cells following induction of G1 replication, which is easily explained by ongoing replication coming from replisomes that had initiated during G1 (Supplementary Fig. 5a).

R1-4

- The paper provides data suggesting that replisomes activated in the absence of CDK activity differ from replisomes activated during normal S-phase. It would be interesting to learn how replisome composition (specifically, the abundance of CMG) in cells undergoing early-onset replication would be affected by the release from alpha factor. For example, if low ratios of MCM10 persist upon release from alpha factor, could faster replication with low-MCM10 replisomes underly the observed DNA damage?

This point is conceptually related to the previous. To address how the composition of S-phase replisomes changes due to prior G1 replication, we used our quantitative MS-based measurement of replisome composition and abundance. We applied it to cells undergoing S-phase replication after prior G1 replication and compared to S-phase replisomes. Interestingly, prior G1 replication does not substantially affect replisome abundance and composition in S-phase (Supplementary Fig. 5c-e). We conclude from this experiment that cellular pathways controlling replisome assembly in S-phase are generally not affected by prior G1 replication. Interestingly, Mcm10 does not follow the general trend and we do see a 4-fold reduction in its association with replisomes also in S-phase. At the moment, it is unclear whether this reflects an increased number of stalled replisomes or whether Mcm10 assembly is defective under those conditions. We aim to investigate this at mechanistic detail in a follow-up project, but discuss a potential contribution to the replication fork collision phenotype in this paper. We furthermore note that despite our progress in being able to quantify replisome composition, we can at this point only provide bulk data showing average replisome composition. We are therefore unable to comment whether protein composition of G1 replisomes changes upon progression into S phase due to their low abundance (< 10% of S phase replisomes).

R1-5

- If the model in Figure 7 is correct, allowing early-onset replication forks to terminate before the release from alpha factor (for example by terminating the overexpression of Dbf4 and Dbf11) should prevent DNA breakage. Can this be tested?

Yes, we agree that reducing the number of fork collisions - either by reducing the number of "remaining G1 forks" or by reducing the number of "S-phase forks" - should suppress the induction of DNA breaks/ DNA damage response.

Unfortunately, this hypothesis cannot be tested directly as it is currently impossible within the experimental system to (i) quickly and tightly shut-off the firing factors *and* (ii) provide sufficient time for all replication forks to terminate. Nonetheless, two lines of experiments in the manuscript provide supporting evidence for this model. First, we show that inhibition of origin firing in S-phase (thereby reducing the number of S-phase forks) largely suppresses the occurrence of DNA damage (Fig. 4e, f). Second, we follow up the idea that successful termination of all replication forks along a chromosome will result in the complete duplication of the affected chromosome. Therefore, we analysed the chromosomal content of survivors of G1 replication. We were struck to observe the occurrence of whole chromosome duplications among the majority of survivors (Fig. 5h, i) We note that complete replication

of an entire chromosome during G1 replication will avoid replication collisions during S phase. Therefore, the high levels of chromosome duplications are in support of the replication collision model.

Minor

R1-6m

- Figure 1B, left panel, bottom row: EdU incorporation is not evident in the asynchronous cell population, although this population should contain S-phase cells. Why?

We apologize that the first version of our manuscript was unclear on this point, but EdU was only added after the induction of G1 replication in this experimental setup. Replication products from the asynchronous population could therefore not be labelled with EdU. We now have clarified this point by removing the asynchronous sample and clearly indicating the point of EdU addition.

Reviewer 2

GENERAL COMMENTS

In this work, Reusswig and colleagues used budding yeast as a model to study the unscheduled replication in the G1-phase and its consequence in the following S-phase cells. The authors engineered genetic systems in budding yeast to generate strains inducing unscheduled replication in the G1-phase, and quantified the composition of replisomes in G1- and S-phase by quantitative proteomics analysis and showed some interesting differences between G1 replisome and S-phase replisome. The authors further showed that G1 replication per se did not trigger cellular checkpoints, while subsequent replication during S-phase of cells with unscheduled replication in G1 lead to over-replication, chromosome breaks and cell cycle arrest. Strikingly, the authors observed a strong strand-biased occurrence of RPA-bound single-stranded DNA in the subsequent S-phase cells and suggest a model in which head-to-tail replication fork collisions between S-phase replication forks and G1 replication forks to explain their observation.

The authors used both molecule biology and genome-wide approach to study the unscheduled replication in budding yeast cells. The authors performed considerable works with engineering numerous yeast strains bypass CDK and DDK control to characterize the unscheduled G1 replication and its consequences. The manuscript is well structured and written. In my opinion, it's a well-performed study, which provides interesting results for researchers working in the field of DNA replication stress and genome instability. However, some control experiments need to be added to draw a solid conclusion.

We thank the reviewer for praising our work as a “well-performed study”, for the insightful comments, and for the constructive criticism, which we have addressed entirely and using new lines of experiments. We are therefore confident that the reviewer will find our paper much improved and suitable for publication.

SPECIFIC COMMENTS:

Major comments:

R2-1

- In several places, the authors indicated that the unscheduled G1 replication progresses slower than canonical S-phase replication, e.g. Page 3, the last paragraph of Introduction, Page 13, Discussion "When compared to replication in S-phase, replication in G1 progressed approximately tenfold slower in bulk.... We show that inefficient replication initiation and inefficient replication elongation both contribute to overall slow replication (Fig. 3)." However, when I checked Fig. 3, it's not clear whether it's only due to the amount of initiation (or ongoing replication forks) lower in G1 than S or it's also due to the replication fork speed is slower in G1 replication. The authors should clarify this point. And also, they might need to perform some additional experiments, such as fibre assay to check replication fork speed to make a clearer and stronger conclusion.

We thank the reviewer for raising this important point. The initial statement indeed referred to the bulk effect of overall DNA synthesis and NOT the speed of individual replisomes. To estimate replisome speed, we now made use of the fact that our quantitative MS approach allowed us to quantify relative abundance of replisomes (~8-fold lower during G1 replication compared to S-phase (Fig. 2d), determined for CDK+DDK bypass). Assuming the firing of 160 origins during early S-phase and

quantifying bulk replication in G1 (Fig. 3d, CDK+DDK bypass), we estimate average replisome speeds of 1.5 Kb/min (S-phase) and 1.25 Kb/min (G1 replication) (see methods section for details on the calculation). We therefore do not want to rule out that in addition to controlling replication initiation, the cell cycle state likely also has an impact on replication elongation. These data are furthermore consistent with the increased G1 replication that we observed after overexpression of SPT21, which has not been implicated in replication initiation itself and presumably promotes replication elongation.

These data do not allow us conclusions about the speed of individual replisomes and the distribution of replisome speeds within a population of replisomes. Therefore, we attempted to measure individual replisome speeds using the DNA fiber assay and initiated a collaboration with Benjamin Pardo at Montpellier University, who is an expert on DNA fiber assays and now a co-author of this study. Due to the pandemic situation, we relied on the transportation of samples from Germany to France. Unfortunately, shipping appears to have resulted in the breakage of DNA fibers (illustrated in Reviewer Figure 2) and therefore the analysis at the level of single replication forks remains inconclusive at this point and will be a subject of future studies.

R2-2

Another point is that: do the transcriptional stage, which might can be different between the G1 and S cells (e.g. higher transcription in G1 than S), might also contribute to the observed difference between G1 and S phase replication?

We thank the reviewer for bringing up this important point. Indeed, we considered the possibility that due to differences between transcriptional programs in G1 and S-phase there might be increased replication-transcription conflicts upon induction of G1 replication, similar to what has been shown for premature S-phase entry in human cells after oncogene induction (e.g. Macheret & Halazonetis 2018, Jones et al. 2013 (Petermann lab)). This point is of particular relevance, as our new transcriptome analysis showed that after G1 replication the transcriptome remained highly similar to the G1 transcriptome (Supplementary Fig. 1c, d).

However, at the moment we do not have indication for the occurrence of replication-transcription conflicts. In contrast, G1 replication induces relatively low levels of DNA damage in G1 (Fig. 4b) and this DNA damage in G1 appears to be largely dependent on over-replication (Supplementary Fig. 4g, h). Therefore, while replication-transcription conflicts remain an intriguing possibility, we can unfortunately not manipulate transcription in the current system of G1 replication, which is intrinsically dependent on active transcription of replication initiation proteins. Therefore, to address this point of the reviewer, we discuss the possibility of replication-transcription conflicts in the revised version of the manuscript (see final paragraph of discussion, p. 17).

R2-3

- Relative to the previous point, in the model suggested in Fig. 7, the authors assume that the S-phase forks go faster to catch up with the G1 forks, which leads to head-to-tail fork collision. However, it's not clear why the forks resulted from the G1 initiation cannot reach the same speed as the forks from the S phase initiation once cells enter S-phase.

Our analysis described above (see response to R2-1) suggests that on average G1 replisomes travel somewhat slower than S replisomes. Moreover, when we blocked replication initiation in S-phase after prior G1 replication (Fig. 4e), total DNA content only increased very slowly during S-phase compared to the previous G1 replication, which is consistent with stalling of G1 replisomes during S-phase.

Systematic differences in speeds of G1 and S replication may, however, not even be necessary to explain fork collisions. Rather, we think that it is possible that stochastic variations in fork speeds lead to tailgating and head-to-tail collisions, which as such is an intrinsic danger of over-replication. To attempt to model the likelihood of these collisions will not only require to measure fork speeds on single molecule level in G1 and S (see response to R2-1), but also to differentiate G1 replisomes/forks from S replisomes/forks, which is currently technically not feasible.

R2-4

- Although I'd like to trust the results of Western blots shown by the authors, without any internal loading control shown together with the figures (Fig 4 and Fig S4), it's really hard to judge the quality of the results. For example, in Fig. 4D, there is a clear variation amongst samples, for instance, Ddc1-Rad9 3h G1, 20, 40 release samples, show a much lower level of Rad53 compared to the same time points of Ddc1 samples. The authors need to provide Western plots results with proper controls.

To address the issue of gel loading, we have now included a loading control in the form of a cross-reactive band in Fig. 4 and Supplementary Fig. 4. We would also like to refer the reviewer to the images of the non-cropped western blots (Supplementary Fig. 8), where cross-reactive bands are clearly visible and ensure equal loading / comparison between lanes.

We would furthermore like to mention that Western Blots against Rad53 and γ H2A were done on the same samples. Therefore, constant levels of Rad53 can be taken as control for protein content of these samples.

R2-5

Also, please change the "control" to "control cells" in all figures (at least the Western plots ones) to avoid any confusion.

We appreciate that our labelling of the different control or G1 replication conditions may have been confusing to the reader. We have therefore introduced graphical labelling to indicate whether CDK and/or DDK controls to origin firing were intact or bypassed. These graphical labels have been clearly defined in Fig. 1a.

R2-6

- I think the asymmetry of RPA-ChIP-seq pattern observed by the authors in the current study is the most striking result. The authors did a good job to show that such an asymmetry pattern is unlikely due to BIR. However, it's not clear at all, how single-ended DSBs resulted from the head-to-tail fork collisions (if it indeed happens, see the previous comment) can explain the observed pattern.

For example, in Fig. 7 B and C, it's not clear which strand is associated with RPA. It will be great to add the RPA on the figures to make it clear. In addition, the authors should make a clearer scheme to explain how single-ended DSBs cause by the head-to-tail fork collisions can explain the observed asymmetry RPA pattern.

We apologize if the first version of our manuscript did not explain well enough how replication fork collisions and single-ended DSBs can generate single-stranded DNA with the observed asymmetry on the scale of entire chromosomes. We have now added additional explanation in the manuscript. Briefly, in contrast to a double-ended DSB, which will by resection generate single-stranded DNA on both sides of the DSB (as we have observed in Peritore et al. (2021) Mol Cell using the same methodology), a resected, single-ended DSB will generate single-stranded DNA asymmetrically. In bulk experiments, we

will, however, only be able to measure asymmetric single-stranded DNA if additionally, these single-ended DSBs occur with a positional bias. Therefore, the asymmetric accumulation of ssDNA was particularly striking to us, because it not only told us that single-ended DSBs/fork collisions were occurring, but that they occurred with positional bias (i.e. preferentially between outward going replication forks, exactly as we observed in our experiments (Fig. 5d and Fig. 6e). Other than BIR (which we excluded experimentally) we do not know of another mechanism how such a pattern of ssDNA could be generated. We think that the positional bias can be easily explained by the relatively low rates of replication initiation that we observed during G1 replication (Fig. 2d) and the preferential location of early-firing origins in the centre of budding yeast chromosomes (see Supplementary Fig. 6i). In this scenario, G1 replication forks directed towards chromosome ends will often lack a “termination partner” and persist into S-phase, where they can lead to fork collisions.

While we think that the head-to-tail fork collision model with positional bias is the simplest model to fully explain our experimental observations, we understand that this conclusion requires some deduction and have therefore included additional model figures in Fig. 7 to explain (i) how head-to-tail collisions results in single-ended DSBs and (ii) how sparse activation of early-firing origins may lead to the positional bias of single-ended DSBs and single-stranded DNA.

We also would like to mention that over-replicating the entire chromosome may be one of the few means to avoid the detriment of unscheduled G1 replication. Strikingly, in new data now included in the manuscript, we find that whole chromosome amplifications are dramatically increased in the few survivors of G1 replication (Fig. 5 h, i).

R2-7

- Page 1, Abstract, “resulted in over-replication and led to chromosome breaks via head-to-tail replication fork collisions that are marked by chromosome-wide, strand-biased occurrence of RPA-bound single-stranded DNA.” Since the head-to-tail replication fork collision is just a model and not demonstrated by any experiment shown in the current manuscript, the authors should not provide such a strong statement in the abstract. The authors might also need to modify their title to better describe what they indeed observe in their study. Or not, the authors need to provide stronger evidence of head-to-tail fork collision to support their statement.

We agree with the reviewer that “head-to-tail replication fork collisions” are an interpretation of our experimental data and with current technology cannot be directly observed in cells. Therefore, and even though we think the model is parsimonious, we have changed abstract and title according to the reviewer’s suggestion.

Minor comments:

R2-8

- At the beginning of the Result section, please explain why the CDK and CDK/DDK bypass will not trigger the premature S-phase entry in the experimental design. Although the authors explain in detail their experimental design in the Method, it’s good to clear explain it too, at least at the beginning of the Result section, which will be useful for the people not working on DNA replication in yeast.

Agreed. We have therefore added an additional description of the experimental system at the beginning of the results chapter. Additionally, we have further verified the system experimentally using

transcriptome analysis, where principal component analysis of transcriptomes showed that G1 replicating cells remain in a “G1-like state” (Supplementary Fig. 1c, d).

R2-9

- In general, the figure legends contain limited information. Please provide more details in the figure legends, which will be very useful for the readers to better understand the figures and do not need to check forward and backward in the text and other figures in order to fully understand.

Agreed. We have expanded figure legends within the limits dictated by journal style.

R2-10

- Fig. 1C, on the Y-axis, it indicates “coverage EdU-IP”, is it a problem of legend? Or not why is it not the same as the other figures using “coverage EdU-IP/Input”?

Also, please provide the Y-axis scale. The same for all other figures showing the genome-wide profiles.

Thank you, the Y-axis labelling in Fig. 1c has now been corrected. For genome-wide profiles, we provide the scale of the y-axes in square brackets at the top and also explain indicate this way of labeling the y-axis scale in the legend.

R2-11

- Fig. 1D, for the cluster 1 early ARS, the EdU-IP in CDK/DDK bypass (Dbf4_Dpb11) is much lower than the CDK (Dpb11) bypass. Can the authors provide an explanation?

In G1 replication conditions, addition of 60 mM HU strongly limits total DNA synthesis. As in the CDK+DDK bypass more origins are activated (compared to CDK bypass only) simultaneously, there is less DNA synthesis at individual origins.

R2-12

- Fig. 2C, G1 replication, please specify which strain was used in the analysis, is it CDK bypass or CDK/DDK bypass. Is there any difference between them?

The quantitative mass spectrometry analysis in Fig. 2 was conducted with the CDK+DDK bypass strain, which we now also indicate in the figure legend. Due to the elaborate nature of the quantitative mass spec analysis of replisomes, we have concentrated on this condition and not compared it to the CDK-only bypass.

R2-13

- Fig. 1 D and Fig. 5E: although, in Fig 1D, the EdU-seq data show a big difference between Dpb11 (only peak around cluster 1 early ARS and not cluster 2) and Dbf4_Dpb11 (peaks on both cluster 1 and cluster 2, and the difference between the two clusters is much smaller), the read asymmetry on RPA-ChIP-seq shown in Fig 5E is very similar for Dpb11 and Dbf4_Dpb11 (e.g. cluster 1 is much stronger than cluster 2). Can the authors comment on that?

The reviewer raises an interesting point, but we do not think that data (Fig. 1d) acquired with HU-treated cells should be used for these kinds of comparisons as the overall amount of DNA synthesis is saturated upon addition of HU (see our response to R2-11). Importantly, if we look at the non-HU data (Fig. 1c), we see less G1 replication in the CDK-only strain compared to the CDK+DDK bypass. However, the

RPA ChIP-seq profile is not too different (Fig. 5e). Why is that? As explained in the results section and our response to R2-6, we think the number and position of remaining G1 replication forks is critical for fork collisions and occurrence of single-stranded DNA. If rates of replication initiation increase (CDK+DDK bypass) this naturally brings with it increased rates of replication termination. Therefore, the number of G1 replication forks carried over to S phase may not be so dramatically different between the two conditions, generating a similar number of replication fork collisions and ssDNA (Fig. 5e). Along the same line, we argue that even dramatically lower G1 replication in the sporadic systems (Fig. 6) generates a significant degree of genome instability as well.

R2-14

- Also, in Fig. 5E, the profiles of RPA-ChIP-seq of 60, 120 and 180 min samples do not seem symmetry around the centres of early ARS, in particular, for the cluster 1. Do the authors have an explanation?

Fig. 5d and Fig. 5e collectively show that chromosomal location (right/left) is the major predictor of the strand asymmetry of single-stranded DNA formed in these experiments (Fig. 5d). In contrast, (and in contrast to what can be observed in S phase (=30 min timepoint), replication origins are not a good predictor of strand-biased RPA accumulation. The observed bias of early origins towards forward strands appears to be generated by the location of early firing origins, which are slightly more frequent on right side of chromosomes (Supplementary Fig. 6i).

R2-15

- Page 9, “Both short and long chromosomes were similarly affected, as shown by RPA read asymmetry scores normalized for chromosome length (Supplementary Fig. 5E).” However, when I checked carefully the Fig. 5H, it seems that for later time points (i.e. 120 min and 180 min), the large chromosomes have higher RPA asymmetry scores than the small ones, and it’s true for both CDK and CDK/DDK bypass. Please add some quantitative analyses to double check this, and if such correlation is confirmed, please further discuss how to explain the results and whether the results agree with their model.

We agree with the reviewer and removed the comparative statement “similarly”. Given the relatively small and fluctuating differences between large and small chromosomes and the small number of datapoints (individual chromosomes) we do not feel confident to make a statement whether there is a significant effect of chromosome length, but this does not rule out that chromosome length may influence for example the frequency of fork collisions.

R2-16

- Fig. 6D, there are clear peaks observed in all 4 samples at a genomic region, which seem to be an early ARS based on Fig. 1E. Am I correct? Can the author comment on it?

Thanks for spotting. The mentioned peaks on chromosome 4 overlap with retrotransposable elements YDR261W-B / YDR261C-D, which are located roughly 70 Kb downstream of the closest early ARS. We have observed such peaks in RPA-ChIP also at other retrotransposable elements across the genome. Importantly, these peaks occur in all G1-arrested cells independent of replication induction, so we consider them as background.

R2-17

- Page 14, “Due to the bi-directional nature of DNA replication, we think the only theoretical solution to this problem is re-duplication of the entire chromosome.” I think the author should be able to test this hypothesis.

Since with the sporadic system (VC-SLD2) that they developed, the authors showed (Fig. S6E,) that the RPA asymmetry was only detected on long yeast chromosomes containing many origins but not the small chromones. We should expect to observe the complete duplication only for the long chromosomes but not the small ones at the late S phase. With the whole genome sequencing data (e.g. the input of the RPA-ChIP-seq) of VC-SLD2 cells already available, the authors should be able to test this hypothesis to check whether it's indeed the case or not.

Such copy number variations are only induced with a low frequency within a population of cells, particularly with the sporadic system and can unfortunately not be measured directly in the input data for RPA-ChIP-seq. To address this important point, we isolated clones surviving the induction of G1 replication (CDK bypass, now added as Fig. 5h, i) and subjected them to whole genome sequencing. Notably, we found a very striking increase in chromosome copy number in many of the surviving cell clones. We view this as a verification of our model that re-duplication of entire chromosomes with resulting aneuploidy is a major pathway to survive G1 replication. Furthermore, we have changed the description of Supplementary Fig. 6 e to “Such asymmetry was detected preferentially on long yeast chromosomes that also contain many origins.”

Reviewer 3

The manuscript “Unscheduled DNA replication in G1 causes genome instability through head-to-tail replication fork collisions” presents an investigation into the molecular details of early initiation of replication in yeast. The authors used flow cytometry, next-generation sequencing, immunoprecipitation-mass spectrometry (IP-MS) and other techniques to get insight into the process. Since my expertise in the context of the manuscript is limited to mass spectrometry and corresponding data analysis, I will be concentrating solely on the IP-MS experiment.

The authors clearly explain the design of the IP-MS experiment: there are triplicate GFP-Psf2 pulldown samples and triplicate controls with untagged Psf2 per condition. The controls provide the baseline for enrichment analysis for each of the three conditions, as depicted on Fig. 2A, 2B, 2C.

The authors use LFQ intensities to calculate IBAQ values of the replisome-associated proteins, which can be helpful for putting the estimates of the absolute amounts on the same scale. Proteins of the GINS complex have very high and nearly constant IBAQ intensities across the pulldowns (Fig. 2D), which is reasonable since the GFP-tag was attached to the Psf2 subunit of GINS. The further discussion is focused on the relative profiles of intensities of the sub-complexes between the S, S-HU and G1 conditions.

I was able to locate 31 replisome-associated proteins from the Fig. 2 and Supplementary Fig. 2 in the proteomic result table from PRIDE. The proteins of interest were identified with at least 8 unique peptides each, providing a reliable basis for the precursor ion-based quantification. The proteins of interest have either none or 1-2 non-unique peptides each, which simplifies the interpretation of the quantitative results in regards to homology/shared peptides. On the technical side, the LFQ intensities in the protein table show largely unimodal distributions with similar medians and means throughout the range, indicating successful intensity normalisation in MaxQuant. These observations suggest that the relative label-free quantification has a good technical basis.

The authors did not provide the scaling of the IBAQ values to absolute protein concentrations. However, this is inconsequential, as the conclusion about the reduced association of DNA polymerase α /primase with G1 replisomes is based solely on the different magnitudes of relative changes between S and G1.

We thank the reviewer for the insightful comments regarding our proteomics experiments and going into detail with our submission of original data. We have addressed all points of the reviewer as follows.

I would like to suggest a few changes in regards to the reporting of the IP-MS experiment:

R3-1

1) Volcano plots on Fig. 2A-C are depicted with the p-values from Student's t-test. It will not change any of the conclusions of the manuscript, but I would suggest that plotting the adjusted p-values after correction for multiple testing would be more appropriate for this data set.

We agree with the reviewer that our initial analysis did allow for an easy judgement whether individual replisome components were enriched compared to the control IP. To address this point, we have added a significance threshold (FDR: 0.05, S0: 1.0) to the Volcano plots of Fig. 2. We decided against plotting adjusted p-values, as replisome components were difficult to annotate in these conditions.

R3-2

2) Furthermore, it can be possible to use the triplicate data for the replisome proteins and perform statistical testing of the strength of divergence between polymerase α /primase and other components. The resulting p-values could complement the discussion of the Fig. 2D.

To address this point, we have determined the significance of changes in replisome composition from G1 to S replication. In Supplementary Table 3, we now show the fold changes of LFQ values for individual replisome proteins as well as significance determined by Student T-test.

R3-3

3) I could not locate a citation regarding IBAQ in the text. Please consider adding a reference, for example, to Schwanhusser et al. in Nature, 2011 , 473(7347):337-42. doi: 10.1038/nature10098, or to another suitable source of your choice.

We agree and have now added the corresponding reference.

REVIEWERS' COMMENTS

Reviewer #1 (Remarks to the Author):

The revision has addressed most of my concerns and the added information is useful and pertinent. The current version would be improved by addressing two remaining issues.

First, the authors have convincingly demonstrated that the DNA synthesis they observed after CDK/DDK bypass exhibits some unique characteristics and differs from early S-phase entry (also commented by reviewer #2). However, I still take issue with the definition of this synthesis as "replication in G1". The clear, textbook, definition of the G1 phase is functional: G1 is the gap-phase (or growth phase) that occurs after mitosis and before the onset of DNA synthesis. This definition does not involve transcriptomics or kinase levels, and clearly relies on the absence of replication. The situation is different in mitosis; mitosis is defined as the stage of cell division and is not defined based on the absence of replication, and therefore, MiDAS is not as confusing. Perhaps consider "early-onset replication" or "replication in a G1-like state".

Second, a very technical suggestion, relates to the difficulties encountered by the authors attempting to perform DNA combing. Although the paper provides sufficient data to stand alone as a publication, I agree with reviewer #2 that measurements of replication fork progression would nicely complement the story. It is unclear from Reviewer Figure 2 which material exactly was shipped to the combing lab; would the difficulties be alleviated by shipping cells, or agarose plugs prior to extraction?

Reviewer #2 (Remarks to the Author):

In this revised manuscript, Reusswig and colleagues have added new data and results to address most of my previous comments. I think the manuscript has been significantly improved and I just have a few minor comments.

- It is good that the authors added the RNA-seq data analysis to support that the cells after CDK and DDK bypass are close to G1 cells than S-phase cells. However, it should be noted that, since the authors used the data of cells throughout S-phase in this analysis, it cannot be excluded that the CDK/DDK bypass cells might have an RNA-seq pattern close to that of early S-phase cells. The authors might need to discuss this point in the discussion.

- On page 7, the authors state that "indicating that lack of histone synthesis constitutes a bottleneck for unscheduled replication in G1". Do their RNA-seq data support this statement?

- In the absence of DNA combing data, the authors quantified their flow cytometry data to estimate the average replication fork speed, which seems to support their model. However, this estimation largely depends on the number of replisomes per cell during S-phase and G1 (CDK/DDK-by pass strain), respectively. Especially, the average speed of G1 replisome is particularly sensitive to it. For example, the current number of replisomes in G1 is estimated to be 40 (therefore fork speed is $50/40=1.25$ Kb/min), whereas if the numbers of replisomes range from 30 to 50, the fork speed will range from 1 to 1.66 Kb/min (the latter is greater than 1.5 Kb/min estimated for the S-phase cells). Therefore, it is important that the authors also include a confidence interval (for both the bulk replication speed and the number of replisomes) in their analysis before drawing any conclusions.

- It is a pity that the authors did not succeed in performing DNA combing to measure replication fork speed, but at least they did try on it. I think it is important to mention in the text (either in the Result section or in the Discussion) that it will be important to perform DNA combing in the future to accurately measure replication fork speed.

- In Figure 1d, it is not clear why the late ARSs show a lower level of EdU-IP/ enrichment ratio in CDK and CDK/DDK by-pass cells than in control cells. I wonder whether it is a normalization issue. Can the author double-check it? And if it is not a problem of normalization, can the authors explain this striking result?

In addition, in the Method, the authors only state that "The tool bigwigCompare was used afterwards to normalize samples to total input DNA". It is hard to get how the normalization has been actually performed. The authors should provide more details on how the normalization has been performed in their analysis.

Reviewer #3 (Remarks to the Author):

I can see that the authors addressed the previously mentioned points regarding the reporting of the proteomic data. Since I do not notice any other methodological issues in the current version, I can recommend the manuscript by Reusswig et al. for publication.

Reviewer 1

The revision has addressed most of my concerns and the added information is useful and pertinent. The current version would be improved by addressing two remaining issues.

We thank the reviewer for the constructive criticism to improve our manuscript and very much appreciate that the reviewer finds the new data useful and pertinent.

First, the authors have convincingly demonstrated that the DNA synthesis they observed after CDK/DDK bypass exhibits some unique characteristics and differs from early S-phase entry (also commented by reviewer #2). However, I still take issue with the definition of this synthesis as “replication in G1”. The clear, textbook, definition of the G1 phase is functional: G1 is the gap-phase (or growth phase) that occurs after mitosis and before the onset of DNA synthesis. This definition does not involve transcriptomics or kinase levels, and clearly relies on the absence of replication. The situation is different in mitosis; mitosis is defined as the stage of cell division and is not defined based on the absence of replication, and therefore, MiDAS is not as confusing. Perhaps consider “early-onset replication” or “replication in a G1-like state”.

We thank the reviewer for further elaborating his/her point on calling the process G1 replication. We certainly agree with the reviewer that the definition of G1 is functional, or rather, historically, by the absence of function (gap-phase). We think, however, that nowadays we know that G1 is much more than a gap, where cells wait for the next replication, but rather a distinct phase of major cellular growth and metabolic activity. As such, we do not think “G1 replication” is an oxymoron, but agree that “replication in a G1-like state” is a good compromise. We have now put forward this definition at the beginning of our results chapter.

Second, a very technical suggestion, relates to the difficulties encountered by the authors attempting to perform DNA combing. Although the paper provides sufficient data to stand alone as a publication, I agree with reviewer #2 that measurements of replication fork progression would nicely complement the story. It is unclear from Reviewer Figure 2 which material exactly was shipped to the combing lab; would the difficulties be alleviated by shipping cells, or agarose plugs prior to extraction?

We agree that it will be highly interesting to study G1 replication rates by DNA combing and have now highlighted this fact in the discussion section of the paper. Regarding the reviewer’s suggestion, since the combing protocol avoids freezing, shipping of cells was not an option and we have indeed prepared and shipped EdU-labelled DNA embedded in agarose plugs, which we thought would have reduced shearing to a minimum.

Reviewer 2

In this revised manuscript, Reusswig and colleagues have added new data and results to address most of my previous comments. I think the manuscript has been significantly improved and I just have a few minor comments.

We thank the reviewer for the insightful comments and suggestions. We are very pleased that the reviewer views the manuscript now as significantly improved.

1 - It is good that the authors added the RNA-seq data analysis to support that the cells after CDK and DDK bypass are close to G1 cells than S-phase cells. However, it should be noted that, since the authors used the data of cells throughout S-phase in this analysis, it cannot be excluded that the CDK/DDK bypass cells might have an RNA-seq pattern close to that of early S-phase cells. The authors might need to discuss this point in the discussion.

We thank the reviewer that the RNA-seq data was important to ensure that despite induction of DNA replication, cells stayed in a "G1-like" state. We have now added a new panel as Supplementary Figure 1d that shows that our S-phase sample (right panel) has a DNA content characteristic of early S phase. Given the rather strong difference of transcriptomes between this control sample and our "G1 replication" sample, we think we can exclude the hypothesis that cells would be in a state of "early S-phase".

2 - On page 7, the authors state that "indicating that lack of histone synthesis constitutes a bottleneck for unscheduled replication in G1". Do their RNA-seq data support this statement?

Yes, we have now added a new panel as Supplementary Figure 3b that reports on histone transcript levels in our RNA-seq dataset. We find that for all core histones transcript levels are 2-5 fold lower in G1 than in S phase. While induction of G1 replication by bypassing CDK/DDK leads to a modest increase in HTA1 and HTB1 transcript levels, overall, we find that transcript levels for all core histones are similar to those observed in G1, but distinct from what we observed in S phase.

3 - In the absence of DNA combing data, the authors quantified their flow cytometry data to estimate the average replication fork speed, which seems to support their model. However, this estimation largely depends on the number of replisomes per cell during S-phase and G1 (CDK/DDK-by pass strain), respectively. Especially, the average speed of G1 replisome is particularly sensitive to it. For example, the current number of replisomes in G1 is estimated to be 40 (therefore fork speed is $50/40=1.25$ Kb/min), whereas if the numbers of replisomes range from 30 to 50, the fork speed will range from 1 to 1.66 Kb/min (the latter is greater than 1.5 Kb/min estimated for the S-phase cells). Therefore, it is important that the authors also include a confidence interval (for both the bulk replication speed and the number of replisomes) in their analysis before drawing any conclusions.

Agreed, we have now added confidence intervals for the bulk replication speed and the proteomics-based quantification of replisomes to our calculation. We now use our mass spectrometry data to determine a confidence interval for replisome abundance and therefore bulk replication speeds.

4 - It is a pity that the authors did not succeed in performing DNA combing to measure replication fork speed, but at least they did try on it. I think it is important to mention in the text (either in the Result

section or in the Discussion) that it will be important to perform DNA combing in the future to accurately measure replication fork speed.

We agree with the reviewer that our analysis based on replisome abundance and replication rates does not replace the need for single molecule replication data obtained by DNA combing. We now further emphasize this in the discussion section of the revised version of our manuscript.

5 - In Figure 1d, it is not clear why the late ARSs show a lower level of EdU-IP/ enrichment ratio in CDK and CDK/DDK by-pass cells than in control cells. I wonder whether it is a normalization issue. Can the author double-check it? And if it is not a problem of normalization, can the authors explain this striking result?

The reviewer is correct that this is a normalization issue. We used the same amount of EdU-labeled DNA from control and G1-replicating cells for construction of the sequencing libraries. Since EdU was incorporated in random locations in control cells and the EdU signal is rather similar to the total DNA signal. In CDK and CDK/DDK-bypass cells, enriching EdU at early ARS loci results in an apparent depletion of EdU from all other genomic loci to result in the same amount of EdU-labeled DNA overall. Scaling the data according to the total amount of EdU-labeling (e.g. as determined by flow cytometry) could potentially adjust for this. However, we decided not to do this since such scaling would also introduce another source of experimental noise and not change the principal finding.

6 - In addition, in the Method, the authors only state that “The tool bigwigCompare was used afterwards to normalize samples to total input DNA”. It is hard to get how the normalization has been actually performed. The authors should provide more details on how the normalization has been performed in their analysis.

Agreed, we apologize for the insufficient description. We have now added more details to the Methods section and also summarize it here: As a first step, we used the tool bamCoverage to separately calculate read counts in 50 bp bins for EdU-labeled and total DNA (only considering uniquely mapping reads in non-repetitive regions) and normalized them for sequencing depth (CPM, counts per million mapped reads). As a second step, we aggregated these data in larger bin sizes (1-2 kb, depending on the analysis) and calculated the ratio of EdU-labeled DNA to total DNA.

Reviewer 3

I can see that the authors addressed the previously mentioned points regarding the reporting of the proteomic data. Since I do not notice any other methodological issues in the current version, I can recommend the manuscript by Reuswig et al. for publication.

We thank reviewer 3 for using her/his expertise to critically review our proteomics experiments and are pleased that she/he recommends our manuscript for publication.